# Earliest infections predict the age distribution of seasonal influenza A cases

Philip Arevalo[1][†]*, Huong Q McLean[2][‡], Edward A Belongia[2], Sarah Cobey[1]

[1]Department of Ecology and Evolutionary Biology, University of Chicago, Chicago, United States; [2]Center for Clinical Epidemiology and Population Health, Marshfield Clinic Research Institute, Marshfield, United States

**Abstract** Seasonal variation in the age distribution of influenza A cases suggests that factors other than age shape susceptibility to medically attended infection. We ask whether these differences can be partly explained by protection conferred by childhood influenza infection, which has lasting impacts on immune responses to influenza and protection against new influenza A subtypes (phenomena known as original antigenic sin and immune imprinting). Fitting a statistical model to data from studies of influenza vaccine effectiveness (VE), we find that primary infection appears to reduce the risk of medically attended infection with that subtype throughout life. This effect is stronger for H1N1 compared to H3N2. Additionally, we find evidence that VE varies with both age and birth year, suggesting that VE is sensitive to early exposures. Our findings may improve estimates of age-specific risk and VE in similarly vaccinated populations and thus improve forecasting and vaccination strategies to combat seasonal influenza.

*For correspondence:
parevalo@uchicago.edu

Present address: †Department of Ecology and Evolution, University of Chicago, Chicago, United States; ‡Center for Clinical Epidemiology and Population Health, Marshfield Clinic Research Institute, Marshfield, United States

## Introduction

Seasonal influenza is a serious public health concern, resulting in approximately 100,000–600,000 hospitalizations and 5000–27,000 deaths per year in the United States despite extensive annual vaccination campaigns (*Reed et al., 2015*). The rapid evolution of the virus to escape preexisting immunity contributes to the relatively high incidence of influenza, including in previously infected older children and adults. How susceptibility arises and changes over time in the host population has been difficult to quantify.

A pathogen's rate of antigenic evolution should affect the mean age of the hosts it infects, and differences in the rate of antigenic evolution have been proposed to explain differences in the age distributions of the two subtypes of influenza A. Compared to H3N2, H1N1 disproportionately infects children (*Gagnon et al., 2018b*; *Caini et al., 2018*; *Khiabanian et al., 2009*). It also evolves antigenically more slowly (*Bedford et al., 2015*). Thus, compared to H3N2, H1N1 is slower to escape immunity in individuals who have experienced prior infection (namely older children and adults), making them less susceptible to reinfection (*Bedford et al., 2015*; *Beauté et al., 2015*; *Caini et al., 2018*; *Khiabanian et al., 2009*). H3N2, in contrast, exhibits well known changes in antigenic phenotype that are expected to drive cases toward adults (*Smith et al., 2004*; *Cobey and Hensley, 2017*). Under this simple model, hosts previously infected with a subtype face equal risk of reinfection (on challenge) with an antigenic variant of that subtype.

The age distributions of influenza cases in exceptional circumstances—pandemics and spillovers of avian influenza—have shown unexpected variation that suggests important effects of prior infection. Excess mortality in some adult cohorts during the 1918 and 2009 H1N1 pandemics correlates with childhood infection with other subtypes (*Gagnon et al., 2013*; *Worobey et al., 2014*; *Gagnon et al., 2018a*). In the post-2009 pandemic period, excess mortality and hospitalization were observed among cohorts first exposed to H2N2 or H3N2 during H1N1pdm-dominated seasons (*Budd et al., 2019*). Similarly, the subtypes circulating in childhood predict individuals' susceptibility

to severe zoonotic infections with avian H5N1 and H7N9, regardless of later exposure to other seasonal subtypes (*Gostic et al., 2016*). These patterns suggest that early influenza infections, and not prior infection per se, strongly shape susceptibility.

Early infections might also affect the protection conferred by influenza vaccination. Foundational work on the theory of original antigenic sin demonstrated that an individual's immune response to influenza vaccination is biased toward antigens similar to those encountered in childhood (*Davenport and Hennessy, 1956*). In some cases, this may result in a narrow antibody response focused on a single epitope (*Davis et al., 2018*). This phenomenon has been suggested to explain an unexpected decrease in vaccine effectiveness (VE) in the middle-aged in the 2015–2016 influenza season (*Skowronski et al., 2017b*; *Flannery et al., 2018*). More generally, it has been hypothesized that biases in immune memory can arise from both past infections and vaccinations and lead to variation in VE that is sensitive to the precise history of exposures (*Smith et al., 1999*; *Skowronski et al., 2017a*).

To measure the effect of early exposures on medically attended infection risk and VE, we fitted statistical models to 3493 PCR-confirmed influenza cases identified through seasonal studies of influenza VE from the 2007–2008 to 2017–2018 seasons in the Marshfield Epidemiologic Study Area (MESA) in Marshfield, Wisconsin (*Belongia et al., 2009*; *Belongia et al., 2011*; *Griffin et al., 2011*; *Treanor et al., 2012*; *Ohmit et al., 2016*; *McLean et al., 2014*; *Gaglani et al., 2016*; *Zimmerman et al., 2016*; *Jackson et al., 2017*; *Flannery et al., 2018*, *Figure 1—figure supplement 1*). Each influenza season, individuals in a defined community cohort were recruited and tested for influenza when seeking outpatient care for acute respiratory infection. Eligibility was restricted to individuals >6 months of age living in MESA who received routine care from the Marshfield Clinic and who presented in an outpatient setting.

We sought to explain the variation in the age distribution of these cases by subtype and over time. Our model predicted the relative number of cases of influenza in each birth year each season as a function of the age structure of the population, age-specific differences in the risk of medically attended influenza A infection, early influenza infection, and vaccination. Despite the extensive antigenic evolution in both subtypes over the study period, we found strong evidence of protection from the subtype to which a birth cohort was likely first infected (the imprinting subtype) and variation in VE by birth cohort.

## Materials and methods

### Study cohort

Cases of PCR-confirmed, medically attended influenza were identified from annual community cohorts based on residency in MESA. MESA is a contiguous geographic area surrounding Marshfield, Wisconsin, where nearly all 61,000 residents receive outpatient and inpatient care from the Marshfield Clinic Health System (*Kieke et al., 2015*). For each influenza season from 2007 to 2008 through 2017–2018, we identified MESA residents >6 months of age who received routine care from the Marshfield Clinic. These individuals were eligible for recruitment into that season's VE study if they sought care for acute respiratory infection. Trained research coordinators recruited patients during clinical encounters in primary care departments, including urgent care, pediatrics, combined internal medicine and pediatrics, internal medicine, and family practice. Patients were enrolled on weekdays, evenings, and weekends when clinical services were provided. Research staff used an electronic appointment system to screen the chief complaints for respiratory or febrile illness. Patients were then approached in-person to assess eligibility based on specific respiratory symptoms and duration of illness. The proportion of patients with medically attended acute respiratory infection (MAARI) who were screened for enrollment varied by season and was largely determined by the volume of patients each day and staffing capacity. Only symptoms and illness duration were used to determine eligibility among those patients who were in the predefined cohort. Patients were also assessed for the presence of medical conditions that put them at high risk for complications from influenza infection, as defined by the Advisory Committee on Immunization Practice (*Smith et al., 2006*). These conditions included cardiovascular disease, diabetes, pulmonary disease, cancer, kidney disease, liver disease, blood disorders, immunosuppressive disorders, metabolic disorders, and neurological/musculoskeletal disorders. We considered subjects vaccinated if they received that

season's influenza vaccine $\geq$14 days before enrollment. For the 2009–2010 season, we only considered receipt of the 2009 monovalent vaccine. The Marshfield Clinic generally does not capture MAARI in nursing facilities with dedicated medical staff, causing undersampling of the oldest age groups. We adjusted for this (Appendix 1: 'Age-specific rates of approachment, enrollment, and nursing home residence').

Each season, recruitment began when influenza activity was detected in the community and usually continued for 12–15 weeks. Symptom eligibility criteria varied by season but included fever/feverishness or cough during most seasons. We retroactively standardized symptom eligibility criteria to only require cough as a symptom. Individuals with illness duration >7 days or presenting in an inpatient (hospital) setting were excluded. After obtaining informed consent, a mid-turbinate swab was obtained for influenza detection. RT-PCR was performed using CDC primers and probes to identify influenza cases, including type and subtype.

## Calculating differences in the age distribution between seasons

We defined the age distribution of each season as the number of cases of the dominant (more common) subtype in each of nine age groups (0–4 year-olds, 5–9 year-olds, 10–14 year-olds, 15–19 year-olds, 20–29 year-olds, 30–39 year-olds, 40–49 year-olds, 50–64 year-olds, and $\geq$65 years old). We excluded the subdominant subtype in each season due to concerns that short-term interference between the subtypes (*Laurie et al., 2015*; *Goldstein et al., 2011*) would affect the age distribution of the rarer subtype. The G-test of independence was used to measure differences in seasons' age distributions.

## Calculating relative risk

To evaluate relative infection risk in different age groups, we measured their relative risk of infection in the first versus second half of each season. This risk is a combination of the chance of infection, conditional on infection (susceptibility), and the rate of contact with infected people. Attack rates should be higher in populations that experience more risk, and therefore these populations should be infected earlier in the epidemic (*Worby et al., 2015*). To calculate relative risk we used an approach similar to *Worby et al., 2015*. We defined the midpoint of each season as the week in which the cumulative number of cases of the dominant subtype among all people exceeded half the total for that season. Weeks before and after this point were assigned to the first and second half of the season, respectively. We assigned each case to one of the five age groups used by *Worby et al., 2015* (0-4 year-olds, 5–17 year-olds, 18–49 year-olds, 50–64 year olds, and $\geq$65 years old). For each age group $g$, we defined relative risk as

$$\frac{C_{\text{first},t,g}}{C_{\text{second},t,g}}, \tag{1}$$

where $C_{\text{first},t,g}$ and $C_{\text{second},t,g}$ are the fraction of cases of the dominant subtype during influenza season $t$ that occurred during the first or second half of the season, respectively. A relative risk >1 indicates that cases in an age group were more likely to occur during the first half of the season.

## Calculating imprinting probabilities

We hypothesized that the subtype of a person's first influenza A infection affects their future susceptibility to that subtype. Testing this hypothesis requires knowing the probability that a person's primary influenza A infection was with a particular subtype. To calculate these probabilities, we emulated the approach of *Gostic et al., 2016*, which assumes these probabilities are determined by a person's year of birth and subsequent exposure to each subtype.

First, we calculated the probability that an individual born in year $y$ received their first influenza A exposure in influenza season $t$. Assuming a constant per-season rate of infection $i_0$, the probability of infection in one season (i.e., the attack rate) is given by

$$Pr(\text{infection in single season}) = 1 - e^{-i_0}. \tag{2}$$

By assuming that the average probability that a naive individual is infected in a single season is 0.28 (*Bodewes et al., 2011*; *Gostic et al., 2016*), we calculated the expected per-season infection rate ($i_0$) as

$$0.28 = 1 - e^{-i_0},$$
$$i_0 = -ln(0.72). \tag{3}$$

However, because the intensity of epidemics varies between seasons ($I_t$, Appendix 1: 'Seasonal intensity') and the fraction of the epidemic experienced by a person depends on their birth year ($\gamma_{y,t}$, Appendix 1: 'Fraction of season experienced'), we considered the time-varying per-season infection rate,

$$i_{y,t} = i_0 I_t \gamma_{y,t}. \tag{4}$$

Therefore, the probability that a naive individual born in year $y$ is infected in season $t$ is

$$a_{y,t} = 1 - e^{-i_{y,t}}. \tag{5}$$

We used $a_{y,t}$ to calculate the fraction of a birth cohort $y$ that received their first influenza A infection in season $t$. Let $U_{y,t}$ be the fraction of people born in year $y$ who were unexposed at the beginning of season $t$ (Appendix 1: 'Calculating the fraction unexposed'). The probability that a person born in year $y$ has their first infection in season $t$ is

$$Pr(\text{first exposure in season } t) = Pr(\text{infected}|\text{unexposed})Pr(\text{unexposed}) = a_{y,t}U_{y,t} \tag{6}$$

We calculated $m_{s,t,y}$, the probability that a person born in year $y$ had their first influenza A infection with subtype $s$ in season $t$, by multiplying $a_{y,t}U_{y,t}$ by the frequency of subtype $s$ in season $t$, $l_{s,t}$ (**Figure 3—figure supplement 1**),

$$m_{s,t,y} = l_{s,t}a_{y,t}U_{y,t}. \tag{7}$$

## Modeling approach

We aimed to predict $p_{s,t,y,v}$, the fraction of cases of subtype $s$ in season $t$ among people born in year $y$ with vaccination status $v$. Our models assume that this is proportional to a combination of the following factors:

1. *Demography*. The age distribution of our study cohort is not static over the study period. All models adjusted for the changing fractions of the population in each birth cohort and season (**Figure 1—figure supplement 2**; Mathematical expressions for model components: 'Demography').
2. *Age-specific effects*. We considered that age itself may be associated with differences in medically attended influenza A infection risk stemming from differences in susceptibility and/or rates of contact with infectious people. Additionally, we expect that age groups may intrinsically differ in their healthcare-seeking behaviors. These factors are inseparable in our data, and all models represent their combined effects with a static age-specific parameter shared by both subtypes that describes the risk of age-specific medically attended influenza A infection (Mathematical expressions for model components: 'Age-specific factors'). We assumed no intrinsic differences in the age-specific virulence of the two subtypes. These age-specific parameters were fitted. We also adjusted for other potential sources of age-specific bias, including age-specific differences in study approachment and enrollment rates (Appendix 1: 'Age-specific rates of approach, enrollment, and nursing home residence').
3. *Imprinting*. We tested several hypotheses of how primary exposures could affect the risk of medically attended infection with H1N1 and H3N2. In each version, we estimated fractional reductions in risk of medically attended H1N1 and H3N2 infection due to primary (i.e., imprinting) exposure to the same type:
   - Subtype-specific imprinting: Influenza has two main antigens, hemagglutinin (HA) and neuraminidase (NA). Imprinting could in theory derive from responses to either or both antigens. Because H1N1 is the only seasonal subtype of influenza with N1, we cannot separate the effects of initial N1 exposure from initial H1 exposure. However, since N2 appears in both H3N2 and H2N2 viruses, we can estimate protection against H3N2 infection from initial N2 exposure separately from protection from initial H3 exposure (Mathematical expressions for model components: 'HA subtype imprinting' and 'N2 imprinting').
   - Group-level imprinting: Influenza A viruses fall into two groups (I and II) corresponding to the two phylogenetic clades of HA. **Gostic et al., 2016** found that primary infection by a

virus belonging to one group protected against severe infection by another subtype in the same group. If group-level imprinting were influential, we would see primary infection with H2N2 conferring protection against H1N1, another group I virus, as well as H1N1 protecting against H1N1, and H3N2 against H3N2. We considered a separate class of models that assumes group-level protection instead of subtype-specific protection (Mathematical expressions for model components: 'HA group imprinting').

4. *Vaccination.* Approximately 45% of the MESA population was vaccinated against influenza each year (*Figure 1—figure supplement 3*; Appendix 1: 'Vaccination coverage'). We estimated cases in vaccinated and unvaccinated individuals of each birth year separately. Naively, we expect that vaccinated individuals should seek medical attention for acute respiratory infection proportionally to the fraction of their cohort vaccinated that season. However, vaccinated individuals may seek medical attention for acute respiratory infection more frequently than non-vaccinees due to correlations between the decision to vaccinate, healthcare-seeking behavior, and underlying medical conditions (*Jackson et al., 2006a*; *Jackson et al., 2006b*; *Belongia et al., 2011*). Indeed, we generally observed higher rates of high-risk medical conditions among vaccinated people compared to unvaccinated people (*Figure 1—figure supplement 4*). We attempted to adjust for this by calculating the fraction of vaccinated people among those who had MAARI and tested negative for influenza (i.e., the test-negative controls, 'Mathematical expressions for model components: Vaccination'). We found that the vaccinated fraction exceeds vaccination coverage for most age groups, suggesting vaccinated individuals are overrepresented among cases for reasons unrelated to influenza (*Figure 1—figure supplement 5*). We also assumed that vaccination is not perfectly effective, and defined VE as the fractional reduction in cases expected in vaccinated compared to unvaccinated individuals after controlling for the effects described above. We estimated subtype-specific VE under five scenarios: (i) constant across age groups and seasons; (ii) constant across age groups but season-specific; (iii) age-specific but constant across seasons; (iv) imprinting-specific; and (v) birth-cohort-specific. We assumed that vaccination affects risk only in the current season, i.e, vaccination in a prior season confers no residual protection (Mathematical expressions for model components: 'Vaccination'; *Ohmit et al., 2014*; *Ohmit et al., 2016*; *Jackson et al., 2017*; *Skowronski et al., 2016*; *Pebody et al., 2013*; *McLean et al., 2018*).

We defined models as specific combinations of the above factors. We tested a set of 10 models by pairing each of the possible implementations of HA imprinting with each implementation of VE (*Figure 1*). Demography, age-specific effects, and N2 imprinting were included in all these models. To test whether more complex models truly improved model fit, we also tested a simple model with constant VE and no effect of imprinting. We evaluated these 11 models by maximum likelihood and compared their performance using the corrected Akaike information criterion (cAIC, 'Model likelihood') and leave-one-out cross-validation.

## Mathematical expressions for model components

### Demography

We expect that the fraction of cases in each birth cohort should be proportional to the underlying demographic birth year distribution of the population. To calculate the demographic birth year distribution, we used MESA-specific data on the age distribution for each season (*Kieke et al., 2015*). Because people $\geq 90$ years old were grouped into a single age class, we estimated the number of people in each age $\geq 90$ years old by assuming a geometric decline in population with age. We converted the age distribution for each season into a distribution by birth year by assigning people of a specific age into the two possible birth years of that age (Appendix 1: 'Birth year distribution of the study population'). Therefore,

$$p_{s,t,y,v} \propto D_{t,y}, \tag{8}$$

where $D_{t,y}$ is the fraction of the population in season $t$ who were born in year $y$.

### Age-specific factors

We modeled intrinsically age-specific differences in medically attended influenza A infection risk and healthcare-seeking behavior by using parameters that represent the relative risk of medically attended influenza A infection in each age group. These parameters combine the effects of underlying age-specific differences in influenza A medically attended infection risk as well as age-specific

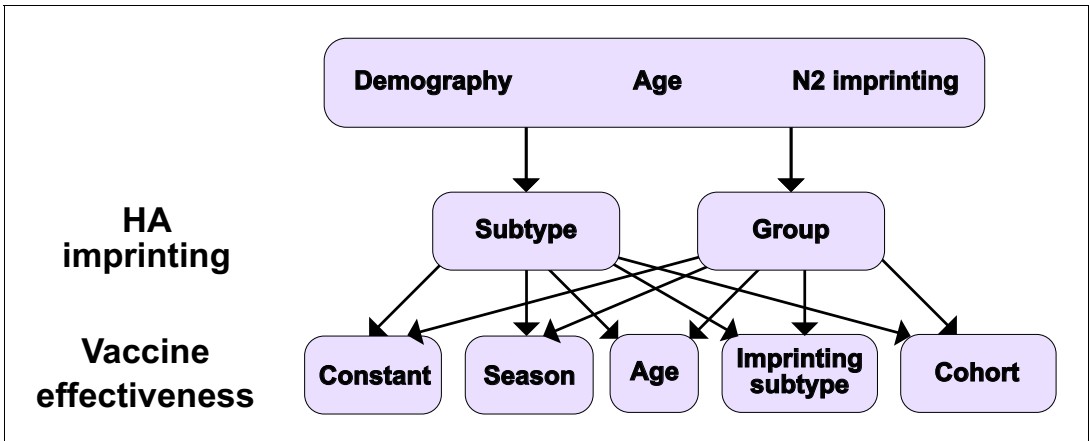

**Figure 1.** Summary of models tested. Ten different models result from considering different combinations of HA imprinting and VE. We also tested one additional model excluding the effects of N2 and HA imprinting (Materials and methods: 'Modeling approach').
The online version of this article includes the following figure supplement(s) for figure 1:

**Figure supplement 1.** Sample collection and final study population.
**Figure supplement 2.** Birth year distribution of population.
**Figure supplement 3.** Vaccination coverage.
**Figure supplement 4.** Age distribution of high-risk medical status.
**Figure supplement 5.** Rate of MAARI in vaccinated and unvaccinated controls.
**Figure supplement 6.** Repeat vaccination by age group and season.
**Figure supplement 7.** Vaccine type received.

differences in healthcare-seeking behavior. We considered the same age groups as before (0–4 year-olds, 5–9 year-olds, 10–14 year-olds, 15–19 year-olds, 20–29 year-olds, 30–39 year-olds, 40–49 year-olds, 50–64 year-olds, and ≥65 years old). We chose 20–29 year-olds as our reference age group. All age groups $g$ aside from 20 to 29 year-olds had an associated parameter ($A_g$) that scaled their risk of medically attended influenza A infection relative to 20–29 year-olds. These parameters can take on any positive value.

Since our models describe the distribution of cases by birth year and not by age, we mapped the age-group-specific parameters ($A_g$) to birth cohorts in each season $t$ ($A_{t,y}$). We considered that each birth cohort has two possible ages in each season ($a1$ and $a2$). Let $G(a)$ be a function that specifies the age group $g$ of a given age $a$. Then $A_{t,y}$, the age-specific relative risk in season $t$ of medically attended influenza A infection for a person born in year $y$, is

$$A_{t,y} = f_{a1,t,y}A_{G(a1)} + f_{a2,t,y}A_{G(a2)}, \tag{9}$$

where $f_{a1,t,y}$ and $f_{a2,t,y}$ are the fractions of birth cohort $y$ who are age $a1$ or $a2$ in influenza season $t$ (Appendix 1: 'Fraction of birth cohort with specific age'), and $A_{G(a1)}$ and $A_{G(a2)}$ are the age-group-specific parameters for $a1$ and $a2$.

Our models also included age-specific approachment rates ($x'_{\text{approach},t,y}$), enrollment rates ($x'_{\text{enroll},t,y,v}$), and nursing home enrollment ($k_{t,y}$) as covariates, all of which bias the age distribution of medically attended influenza infections (Appendix 1: 'Age-specific rates of approach, enrollment, and nursing home residence'). The combination of estimated age-specific effects and age-specific covariates was modeled as

$$p_{s,t,y,v} \propto A_{t,y}x'_{\text{approach},t,y}x'_{\text{enroll},t,y,v}(1 - k_{t,y}). \tag{10}$$

## HA subtype imprinting
We considered that imprinting to HA reduces a birth cohort's risk of future infection from the same HA subtype. Therefore,

$$p_{s,t,y,v} \propto 1 - h_s m_{s,t,y}, \tag{11}$$

where $h_s$ is the strength of HA imprinting for subtype $s$ and $m_{s,t,y}$ is the imprinting probability in season $t$ of birth cohort $y$ to subtype $s$ ('Calculating imprinting probabilities').

## HA group imprinting

We considered that imprinting to HA reduces a birth cohort's risk of future infection with viruses from the same HA group. Therefore,

$$p_{\text{H1N1},t,y,v} \propto 1 - g_1(m_{\text{H1N1},t,y} + m_{\text{H2N2},t,y}), \tag{12}$$

$$p_{\text{H3N2},t,y,v} \propto 1 - g_2 m_{\text{H3N2},t,y}, \tag{13}$$

where $g_1$ is the strength of HA imprinting for group one viruses; $g_2$ is the strength of HA imprinting for group two viruses; and $m_{\text{H1N1},t,y}$, $m_{\text{H2N2},t,y}$, and $m_{\text{H3N2},t,y}$ are the imprinting probabilities in season $t$ of birth cohort $y$ to H1N1, H2N2, and H3N2.

## N2 imprinting

We considered that imprinting to N2 reduces a birth cohort's risk of H3N2 infection. Therefore,

$$p_{\text{H3N2},t,y,v} \propto 1 - n_m(m_{\text{H3N2},t,y} + m_{\text{H2N2},t,y}), \tag{14}$$

where $n_m$ is the strength of N2 imprinting, and $m_{\text{H3N2},t,y}$ and $m_{\text{H2N2},t,y}$ are the imprinting probabilities of birth cohort $y$ in season $t$ to H3N2 and H2N2.

## Vaccination

We assumed that vaccination decreases the risk of medically attended infection. However, vaccinated individuals may seek healthcare for symptomatic influenza at a different rate than unvaccinated individuals. Moreover, because vaccines are routinely recommended for individuals with underlying health conditions, pre-existing susceptibility to MAARI among vaccinated individuals may also differ from unvaccinated individuals. Let $R_{t,g}$ represent the fraction of vaccinated individuals in age group $g$ in season $t$ that present with MAARI. We use test-negative controls to estimate this as

$$R_{t,g} = \frac{v_{t,g}^-}{u_{t,g}^- + v_{t,g}^-}, \tag{15}$$

where $v_{t,g}^-$ and $u_{t,g}^-$ are the number of vaccinated or unvaccinated individuals born in year $g$ presenting with MAARI and testing negative for influenza in season $t$. We converted $R_{t,g}$ to $R_{t,y}$ (i.e., to a covariate indexed by birth cohort) using the same method described in 'Age-specific factors.' We tested five different VE schemes: subtype-specific VE that remained constant across seasons and cohorts (two parameters), subtype-specific VE that varied between the age groups described above (18 parameters), VE that varied between seasons (12 parameters), VE for each possible imprinting subtype (six parameters), and birth-cohort-specific VE (18 parameters). These VE parameters ($V$) reduced the probability of medically attended influenza A infection among vaccinated individuals in a birth cohort, i.e,

$$p_{s,t,y,\text{vac.}} \propto R_{t,y}(1 - V) \tag{16}$$

$$p_{s,t,y,\text{unvac.}} \propto (1 - R_{t,y}), \tag{17}$$

where $V$ depends on the specific implementation of VE used.

Constant VE only varies with the infecting subtype, thus

$$V = v_s. \tag{18}$$

Season-specific VE varies with subtype and season, thus

$$V = v_{s,t}. \tag{19}$$

For age-specific VE, we used the same age classes described above for 'Age-specific factors' but

did not consider a reference age class, so that each age group had an associated VE for each subtype. We used these age-specific VE parameters to calculate the VE against subtype $s$ in birth cohort $y$ during season $t$ using the same procedure described in 'Age-specific factors' (*Equation 9*). Therefore,

$$V = f_{a1,t,y} v_{G(a1),s} + f_{a2,t,y} v_{G(a2),s}, \tag{20}$$

where $v_{G(a1),s}$ and $v_{G(a2),s}$ are age-specific VE parameters for $a1$ and $a2$.

For imprinting-specific VE, we used the imprinting probabilities for each birth cohort described in 'Calculating imprinting probabilities' to scale V such that

$$V = 1 - \prod_{z \in \{H1N1, H2N2, H3N2\}} (1 - v_{s,z} m_{z,t,y}), \tag{21}$$

where $v_{s,z}$ is the VE among people imprinted to subtype $z$ against infection by dominant subtype $s$, and $m_{z,t,y}$ is the imprinting probability for subtype $z$ in season $t$ for birth cohort $y$.

For birth-cohort-specific VE, we defined nine birth cohorts corresponding to the nine age groups we used for the 2017–2018 season: 1918–1952, 1953–1967, 1968–1977, 1978–1987, 1988–1997, 1998–2002, 2003–2007, 2008–2012, and 2013–2017. Let $Q(y)$ be the birth cohort of people born in year $y$. Then

$$V = v_{Q(y),s}, \tag{22}$$

where $v_{Q(y),s}$ is the VE among people in cohort $Q(y)$ against infection by dominant subtype $s$.

## Model likelihood

Recall that our aim is to predict $p_{s,t,y,v}$, the fraction of all PCR-confirmed influenza cases of dominant subtype $s$ in influenza season $t$ among people born in year $y$ with vaccination status $v$. These fractions can also be interpreted as multinomial parameters that describe the probability that in season $t$, a medically attended influenza infection of subtype $s$ occurs among people born in year $y$ with vaccination status $v$. Each model $M$ assumes that $p_{s,t,y,v}$ is proportional to a collection of model components $j$ described above (demography, age, imprinting, and vaccination). Thus,

$$p_{M,s,t,y,v} \propto \prod_j \phi_{M,j} \eta_{j,s,t,y,v}, \tag{23}$$

where $p_{M,s,t,y,v}$ is a multinomial probability under model $M$, $\phi_{M,j}$ indicates whether model $M$ contains component $j$, and $\eta_{j,s,t,y,v}$ is the mathematical expression for model component $j$ given $s$, $t$, $y$, and $v$ (e.g., for HA subtype imprinting, $\eta_{j,s,t,y,v} = 1 - h_s m_{s,t,y}$).

To obtain proper multinomial probabilities, we calculated a normalizing constant for each season $t$ such that all probabilities in that season sum to 1. For convenience, let $p'_{M,s,t,y,v} = \prod_j \phi_{M,j} \eta_{j,s,t,y,v}$ be the unnormalized multinomial probability for model $M$. Then for a specific season $t$, the normalized multinomial probability is

$$p_{M,s,t,y,v} = \frac{p'_{M,s,t,y,v}}{\sum_{y'=1918}^{y_{\max,t}} p'_{M,s,t,y',\text{unvac.}} + \sum_{y'=1918}^{y_{\max,t}} p'_{M,s,t,y',\text{vac.}}}. \tag{24}$$

where $y_{\max,t}$ is the maximum birth year possible for a specific season $t$.

To calculate the likelihood of a given model, we used the multinomial probabilities and the observed birth year distribution of cases. Let $n_{s,t,y,v}$ be the number of PCR-confirmed cases of dominant subtype $s$ in influenza season $t$ among people born in year $y$ with vaccination status $v$. The total number of PCR-confirmed cases of dominant subtype $s$ in season $t$ is

$$N_{s,t} = \sum_{y=1918}^{y_{\max,t}} n_{s,t,y,\text{unvac.}} + \sum_{y=1918}^{y_{\max,t}} n_{s,t,y,\text{vac.}}. \tag{25}$$

For models fitted to a restricted set of ages, we limited the cases for each season to the birth cohorts that were guaranteed to meet the age requirements in that season.

Then, the likelihood of model $M$ in season $t$ is given by the multinomial likelihood,

$$\mathcal{L}_{M,t} = \frac{N_{s,t}! p_{M,s,t,1918,\text{unvac.}}^{n_{s,t,1918,\text{unvac.}}} p_{M,s,t,1918,\text{vac.}}^{n_{s,t,1918,\text{vac.}}} \cdots p_{M,s,t,y_{\max,t},\text{unvac.}}^{n_{s,t,y_{\max,t},\text{unvac.}}} p_{M,s,t,y_{\max,t},\text{vac.}}^{n_{s,t,y_{\max,t},\text{vac.}}}}{n_{s,t,1918,\text{unvac.}}! n_{s,t,1918,\text{vac.}}! \cdots n_{s,t,y_{\max,t},\text{unvac.}}! n_{s,t,y_{\max,t},\text{vac.}}!}, \tag{26}$$

Finally, the full model likelihood for model $M$ over all observed seasons is

$$\mathcal{L}_M = \prod_{t=2007-2008}^{2017-2018} \mathcal{L}_{M,t}. \tag{27}$$

We fitted the model to case data using the L-BFGS-B algorithm implemented in the R package *optimx*. We estimated 95% confidence intervals for parameters of the best-fitting model by evaluating likelihood profiles at 14 evenly spaced points and interpolating the entire profile using a smoothing spline.

## Results

### The age distribution of cases varies between seasons and subtypes

The age distribution of cases varies between subtypes. The relative burden of cases is consistently higher in people ≥65 years old during H3N2-dominated seasons compared to H1N1-dominated seasons (*Figure 2*). The age distribution tends to vary more between subtypes than within either over time (*Figure 2—figure supplement 1*, off-diagonal quadrants). This is consistent with recent work showing that the ratios of H3N2 to H1N1 cases differ between age groups (*Gagnon et al., 2018b*).

The age distribution also varies within subtypes over time (*Figure 2—figure supplement 1*, diagonal quadrants). The seven H3N2-dominated seasons display three types of age distributions (*Figure 2—figure supplement 1*, clusters of lighter-colored cells in the upper left-hand quadrant), and two correspond to major antigenic clusters (2007–2008, *Fonville et al., 2016*; 2010–2012, *Ann et al., 2012*). These differences sometimes coincide with significant shifts in the age distribution between seasons. For instance, the highest fraction of H3N2 cases occurs in 20–29 year olds in the 2007–2008 season, but this age group has the lowest fraction of cases in the next H3N2-dominated season (2010–2011, *Figure 2C*). In H1N1, the shift from seasonal to pandemic strains is associated with large changes in the age distribution (*Figure 2—figure supplement 1*, lower right-hand quadrant).

We found further evidence that age groups differed in their susceptibility across seasons by examining the relative risk of infection during the first versus second half of each epidemic period (Materials and methods: 'Calculating relative risk'). Individuals at greater risk of infection should be infected disproportionately early rather than late in an epidemic (*Worby et al., 2015*). We confirmed that an age group's relative risk correlates with the fraction of cases within that age group in the same season (Pearson's r = 0.58, 95% CI 0.38–0.73; *Figure 2—figure supplement 2A*; Appendix 1: 'Correlation of relative risk and fraction of cases'). This trend is evident for H1N1 (Pearson's r = 0.73, 95% CI 0.45–0.88; *Figure 2—figure supplement 2A*) and H3N2 seasons separately (Pearson's r = 0.52, 95% CI 0.30–0.69; *Figure 2—figure supplement 2A*). The positive correlation in all seasons is robust to undersampling of cases at the start and end of seasons (Appendix 1: 'Sensitivity to sampling effort', *Figure 2—figure supplement 2B*). This provides supporting evidence that the different numbers of cases in each age group reflect underlying differences in infection risk.

Just as the age distribution of cases varies over time, the age groups with high relative risks of infection change over time. If people contact one another similarly from one season to the next, these shifting relative risks imply that age groups' relative susceptibilities change over time. For instance, 5–17 year olds had the highest relative risk of early infection in the 2008–2009 season, whereas 50–64 year-olds had the highest relative risk in the 2013–2014 season (*Figure 2—figure supplement 3*). Relative risks in MESA vary more than national estimates, which show that 5–17 year-olds had the highest relative risk in all but one season from the 2009 pandemic to 2013–2014 (*Worby et al., 2015*). These differences may partly be due to the fact that our measurements of relative risk use outpatient visits, whereas the national estimates use hospitalizations.

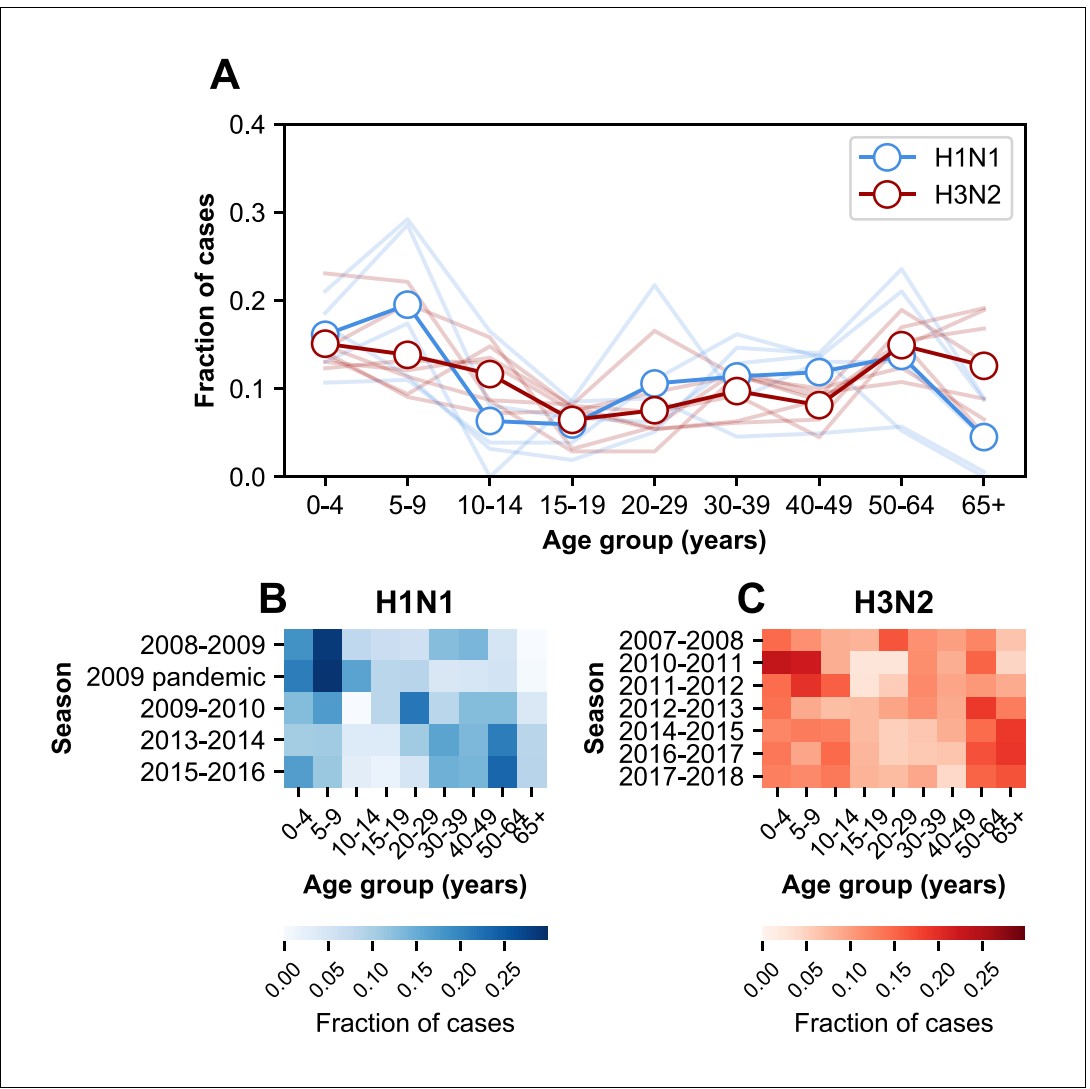

**Figure 2.** The age distribution of cases. (**A**) The age distributions of cases from the 2007–2008 through the 2017–2018 influenza seasons in MESA. Dark lines with open circles indicate the average fraction of cases in each age group. Lighter-colored lines show the age distribution for individual seasons. (**B**) The age distribution of cases in H1N1-dominated seasons. (**C**) The age distribution of cases in H3N2-dominated seasons.

The online version of this article includes the following figure supplement(s) for figure 2:

**Figure supplement 1.** Statistical analysis of age distribution of cases.
**Figure supplement 2.** Correlation of relative risk and fraction of cases within an age group.
**Figure supplement 3.** Relative risk among different age groups across seasons.

Taken together, these findings suggest that the risk of influenza infection is not a simple function of age alone. Other factors, such as past influenza infections and vaccination, might explain the changing age distributions of cases in time.

## Imprinting probabilities of age groups change over time

We hypothesized that variation in the age distribution of cases could be explained by the aging of birth cohorts with similar early exposure histories. This would cause the early exposure history of an age group, and thus potentially its susceptibility, to change in time. To calculate the probability that people in a particular age group had their first influenza A infection with a particular subtype, we adapted the approach from *Gostic et al., 2016*. Briefly, we calculated the probability that an individual born in a specific year had a primary infection with H1N1, H2N2, or H3N2 using data on relative

epidemic sizes and the frequencies of circulating subtypes (*Figure 3—figure supplement 1*; Materials and methods: 'Calculating imprinting probabilities').

As expected, age groups' early exposures are not static and change over time (*Figure 3*). Older people nonetheless tend to be imprinted to H1N1 or H2N2, whereas younger people have higher probabilities of imprinting to H3N2. The effects of the 2009 H1N1 pandemic are evident in the three youngest age groups as a transient increase (from 2009 to approximately 2013) in their H1N1 imprinting probability. These imprinting probabilities are relatively well-constrained even after for accounting for uncertainty in epidemic size (*Figure 3—figure supplement 2*; Appendix 1: 'Sensitivity to uncertainty in ILI and the frequency of influenza A').

## Age-specific differences in medically attended influenza A infection risk affect epidemic patterns

We fitted models to estimate the underlying effects of age, early infections, and vaccination on the age distributions of cases. As expected, the cases reveal age-specific differences in the risk of medically attended influenza A infection (*Figure 4*; *Figure 4—figure supplement 1*; *Appendix 2—table 1*). This risk is roughly threefold higher among children <4 years old compared to adults 20–29 years old, after adjusting for other effects (*Figure 4*). The decline in risk through middle age is generally consistent with attack rates estimated from serology (*Monto et al., 1985*; *Bodewes et al., 2011*; *Wu et al., 2010*; *Huang et al., 2019*) and clinical infections (*Wu et al., 2017*). We recently observed smaller differences in the attack rates of school-aged children and their parents when estimating

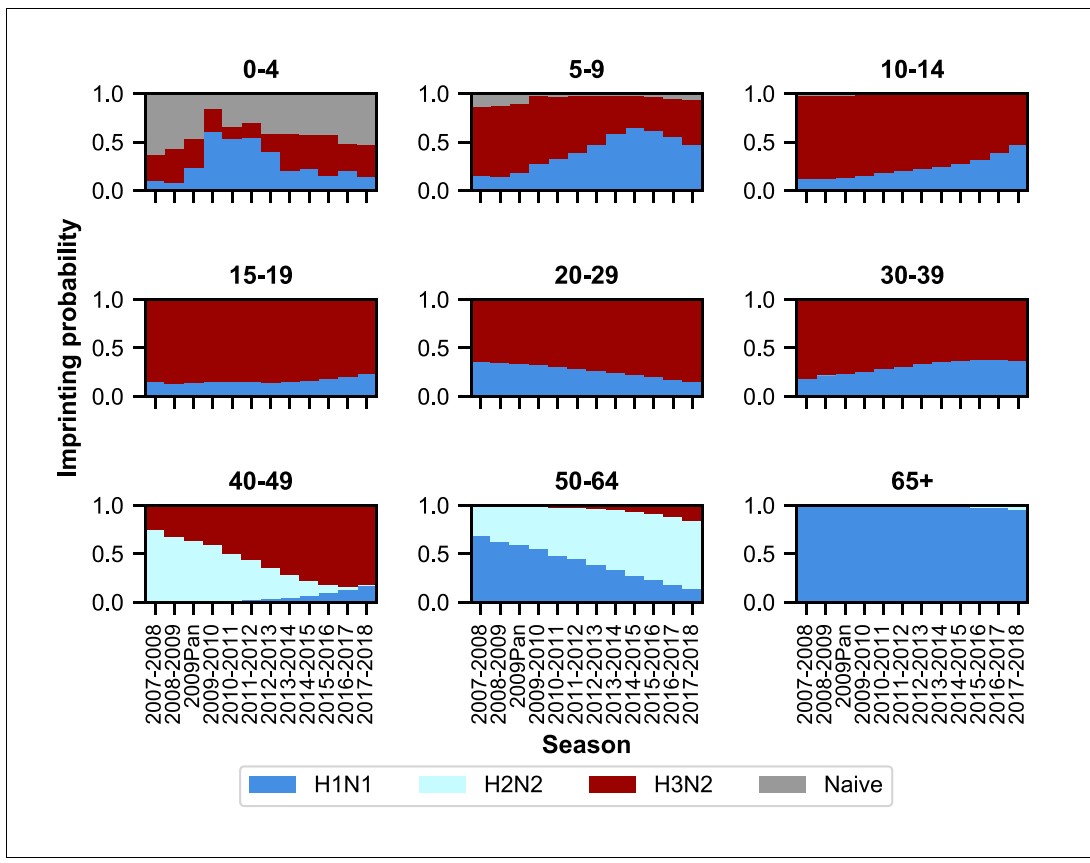

**Figure 3.** Imprinting probabilities by age group across seasons. Each panel shows the imprinting probabilities of an age group from the 2007–2008 season through the 2017–2018 season. The color of each bar corresponds to the imprinting subtype or naive individuals, who have not yet been infected.
The online version of this article includes the following figure supplement(s) for figure 3:

**Figure supplement 1.** Intensity and subtype frequencies of influenza A.
**Figure supplement 2.** Imprinting probabilities with random sampling of seasonal intensity.

infections serologically (**Ranjeva et al., 2019**). We hypothesize that the attack rates estimated from clinical infections might show larger differences by age due to age-related changes in infection severity and healthcare-seeking behavior. Indeed, rates of healthcare-seeking behavior have been shown to decline with age before rising in adults ≥65 years old (**Biggerstaff et al., 2014**; **Brooks-Pollock et al., 2011**; **Van Cauteren et al., 2012**), consistent with our results. Finally, the increased risk of medically attended influenza A infection among people ≥65 years old compared to other adults may be related to the increasing prevalence of high-risk medical conditions with age (**Figure 1—figure supplement 4**).

## Initial infection confers long-lasting, subtype-specific protection against future infection

Our best-fitting model supports subtype-specific imprinting for H1N1 and H3N2 (**Figure 5**, top row; **Appendix 2—table 1**). This model also provides the best predictive power compared to other models in a leave-one-out cross-validation analysis (**Figure 5—figure supplement 1**; **Figure 5—figure supplement 2**; Appendix 1: 'Evaluation of predictive power'). The risk of future medically attended infection by H1N1 is reduced by 66% (95% CI 53–77%) in people imprinted to H1N1, whereas the risk of future medically attended infection by H3N2 is reduced by 33% (95% CI 17–46%) in people imprinted to H3N2. We found no evidence of a protective effect from imprinting to N2 (0%, 95% CI 0–7%). These estimates of imprinting protection are insensitive to:

- uncertainty in imprinting probabilities due to uncertainty in past epidemic sizes (**Figure 3—figure supplement 2**; Appendix 1: 'Sensitivity to uncertainty in ILI and the frequency of influenza A'; **Appendix 2—table 3**),
- choice of age groups for medically attended influenza A infection risk and VE (Appendix 1: 'Sensitivity to age groups'; **Appendix 2—table 4**), and
- undersampling of influenza cases in some seasons (**Figure 5—figure supplement 3**).

In theory, the estimated protective effects of imprinting could be influenced by cross-protection rather than the impact of first infection per se. Because first infections are also recent infections in children, we reasoned that the observed imprinting effects might arise from confounding with recent infections in these ages. Based on an estimated 7 year half-life of homologous protection after H1N1pdm infection in children (**Ranjeva et al., 2019**) and the fact that most children experience primary influenza A infection by 5 years of age (**Bodewes et al., 2011**), we reasoned that excluding

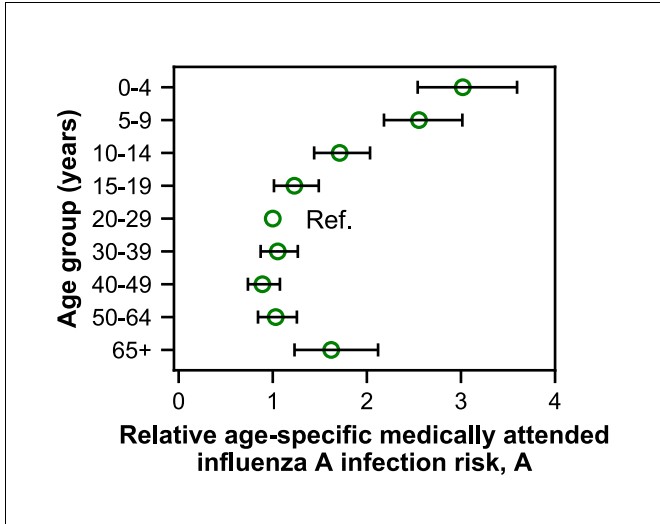

**Figure 4.** Estimates of relative age-specific medically attended influenza infection risk. Open circles represent the maximum likelihood estimates of parameters describing age-specific differences in the relative risk of medically attended influenza A infection. Lines show the 95% confidence interval.

The online version of this article includes the following figure supplement(s) for figure 4:

**Figure supplement 1.** Ranking of models fitted to all ages.

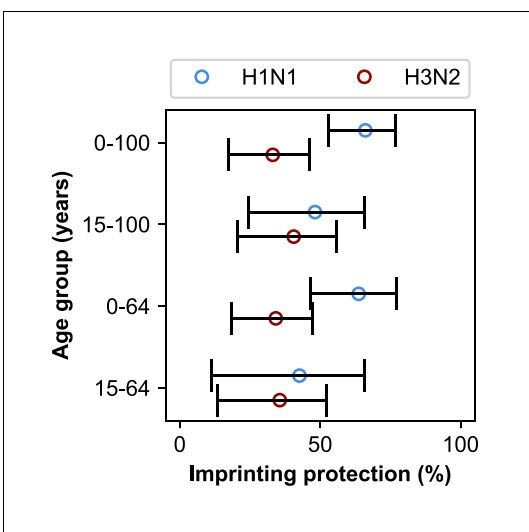

**Figure 5.** Estimates of imprinting strength. Imprinting is more protective against H1N1 infection than H3N2 infection. Open circles represent the maximum likelihood estimates of imprinting parameters from the model including HA subtype imprinting and age-specific VE fitted to the indicated age group (y-axis). Black lines show 95% confidence intervals.

The online version of this article includes the following figure supplement(s) for figure 5:

**Figure supplement 1.** Ranking of models by predictive power.
**Figure supplement 2.** Model performance on excluded seasons.
**Figure supplement 3.** Cases per sampling day.
**Figure supplement 4.** Estimates of imprinting protection with added simulated cases.
**Figure supplement 5.** Correlation of excess cases between seasons.

children <15 years old would diminish the impact of protection from recent infection on our results. When we excluded the youngest age groups, our estimates of H1N1 imprinting protection decreased while H3N2 imprinting protection increased (*Figure 5*, second row). However, initial infection by H1N1 was still more protective than initial infection by H3N2, both imprinting effects remained positive, and there was no significant change in the values of other estimated parameters (*Appendix 2—table 1* and *Appendix 2—table 2*).

The effects of recent infection should also manifest in the difference between the observed and estimated numbers of cases (i.e., the excess cases, Appendix 1: 'Calculating excess cases'), since unlike typical transmission models, our model does not take prior-season infections into account when estimating cases for the current season. More infections in a birth cohort in one season should reduce susceptibility in that birth cohort at the start of the next season. We thus expect that excess cases in one season will be followed by missing cases in the next season dominated by that subtype (i.e., a negative correlation in excess cases). Instead, we observed that excess cases for each birth cohort are weakly positively correlated from season to season, suggesting that immunity from recent infections is not a major driver of temporal variation in the age distribution of cases (*Figure 5—figure supplement 5*).

Since older adults have the highest probability of primary infection with H1N1, we also reasoned that older adults might disproportionately drive the strong protection from H1N1 imprinting we observe. People born before 1947 were likely exposed to H1N1 strains that are antigenically similar to the post-pandemic H1N1 strains that comprise most of our H1N1 infection data (*Manicassamy et al., 2010*; *O'Donnell et al., 2012*), creating the possibility that strain-specific cross-immunity drives the pattern we attribute to subtype-specific imprinting. These people nearly all fall into the ≥65 year-old age group in the study period. The study also underenrolled medically attended infections among people in nursing facilities, which would artificially lower the case count in this age group and may affect estimates of imprinting protection. Therefore, we excluded adults ≥65 years old and refitted our models. Excluding the oldest adults does not significantly change

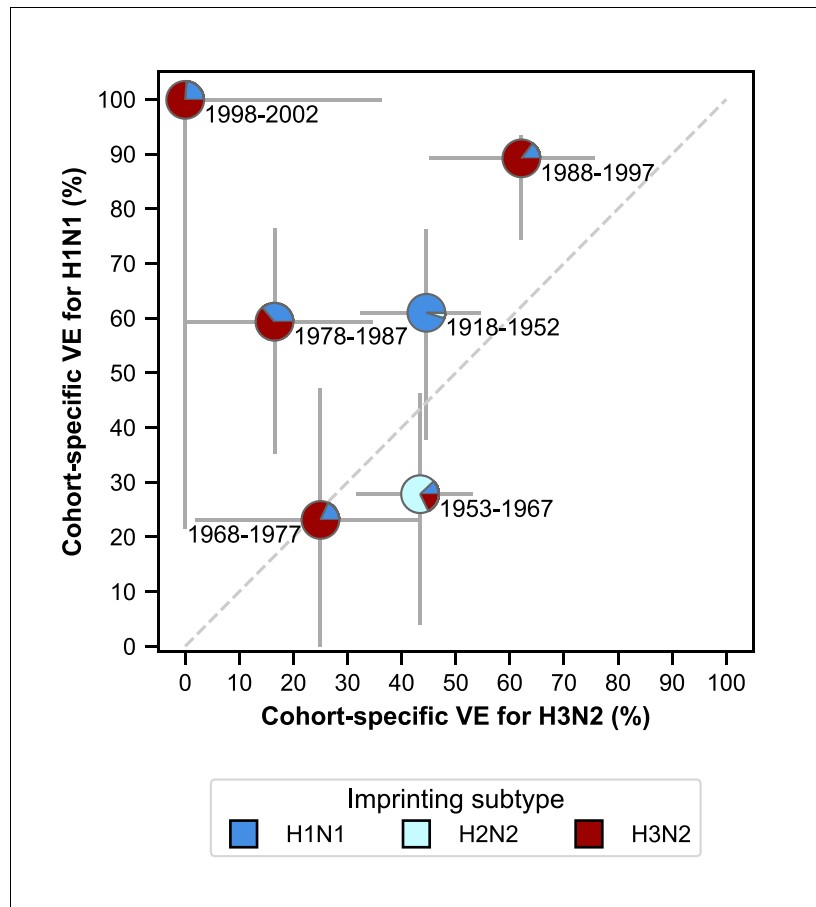

**Figure 6.** Estimates of birth-cohort-specific VE. Birth-cohort-specific VE differs significantly between subtypes and birth cohorts. The location of each pie chart represents the H3N2 (x-axis) and H1N1 (y-axis) VE estimates for a birth cohort (indicated by text) obtained from our model fitted to people ≥15 years old. Pie charts are colored by the probability of first infection by each subtype (i.e, imprinting probability). 95% confidence intervals of the VE estimates are indicated by light grey solid lines. The dashed grey line shows the diagonal where the VE estimate for H1N1 is equal to the VE estimate for H3N2.

The online version of this article includes the following figure supplement(s) for figure 6:

**Figure supplement 1.** Ranking of models fitted to people ≥15 years old.

**Figure supplement 2.** Excess cases for models using birth-cohort-specific VE and age-specific VE.

estimated imprinting protection or other parameters (*Appendix 2—table 1* and *Appendix 2—table 2*).

When we exclude both the youngest and oldest age groups, initial infections by H1N1 and H3N2 have similar protective effects (*Figure 5*, bottom row). This shows that the combined effects of cross-protection in both the youngest and oldest individuals contribute to the signal of imprinting protection we observe, but they are not its sole drivers.

## VE varies by birth cohort in older children and adults

The best-fitting model includes age-specific VE (*Figure 4—figure supplement 1*; *Appendix 2—table 2*). While serological responses to influenza vaccination are weakest in the young (*Englund et al., 2005*; *Neuzil et al., 2006*) and old (*Lee et al., 2018*; *DiazGranados et al., 2014*), it is unclear what age-related factors would drive variation in VE in other age groups. We hypothesized that VE in these ages varies with early exposure history, which correlates with birth year, rather than age.

To test this hypothesis, we fitted a model with birth-cohort-specific VE to the cases, excluding either children <15 years old or adults ≥65 years old. We chose birth cohorts that corresponded to

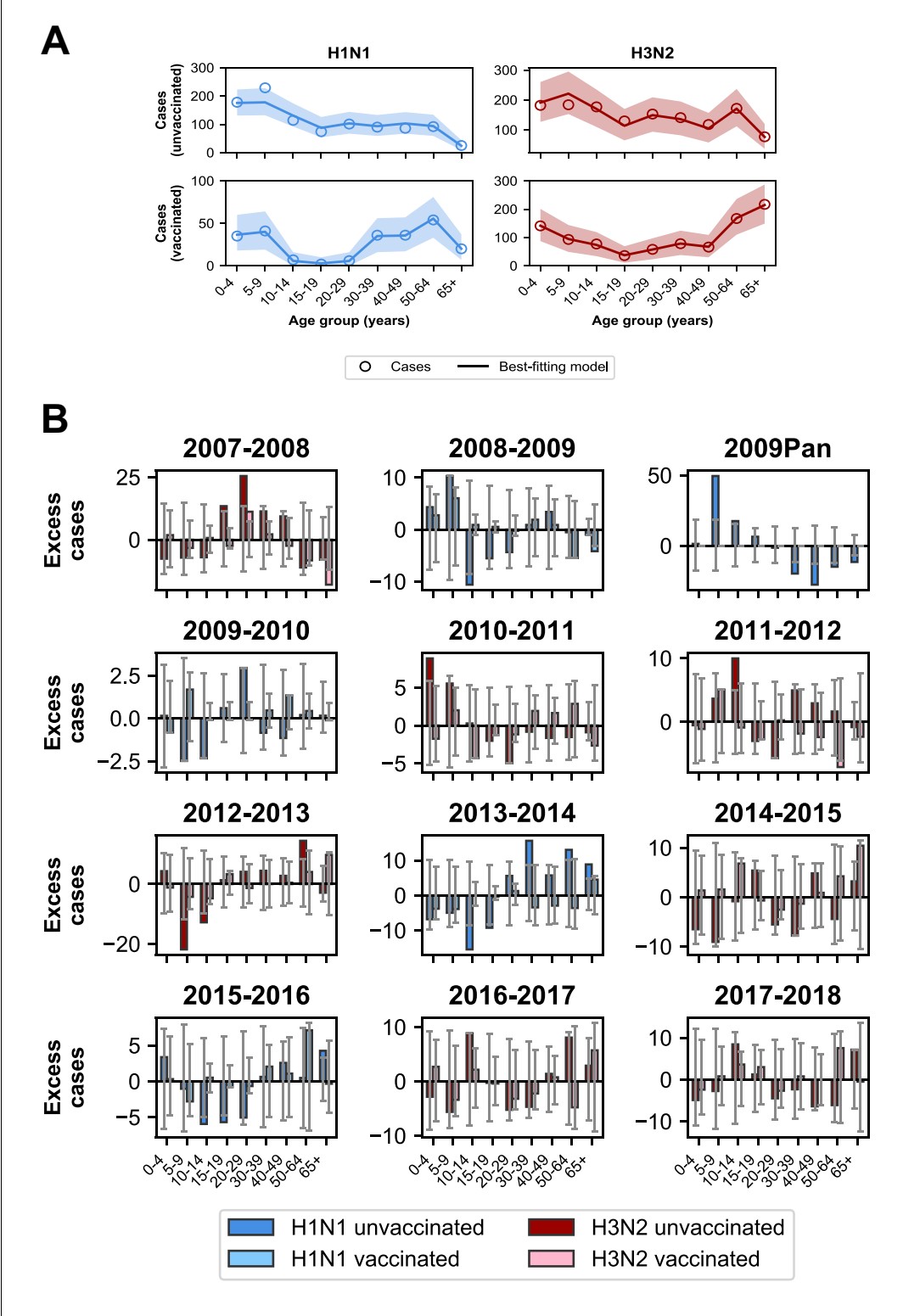

**Figure 7.** Model predictions compared to observed case counts. (**A**) The model including age-specific VE and subtype-specific HA imprinting accurately predicts the overall age distribution of cases across seasons and age groups. Each row depicts the age distribution of cases among unvaccinated (top) and vaccinated (bottom) individuals over all sampled seasons (2007–2008 through 2017–2018). Each column indicates H1N1 cases (left, blue) and H3N2 cases (right, red). Open circles represent observed cases, solid lines represent the predicted number of cases from the best-fitting model, the shaded area represents the 95% prediction interval of the best-fitting model. (**B**) Excess cases of dominant subtype for each season. Excess

*Figure 7 continued on next page*

*Figure 7 continued*

cases are defined as the predicted number of cases from the best-fitting model - observed cases (Appendix 1: 'Calculating excess cases'). Each panel shows the excess cases of the dominant subtype for each season for each age group among unvaccinated (dark bars) and vaccinated (light bars) individuals. Grey error bars show the 95% prediction interval.

the age groups of the original model in 2017–2018 (Materials and methods: 'Vaccination'), keeping the number of parameters the same (e.g., VE in the 20–29 age group became VE in the 1988–1997 birth year cohort). We find that age-specific VE still outperforms all other models after we exclude the oldest age group (≥65 years old). In contrast, birth-cohort-specific VE performs better when we exclude children <15 years old (*Figure 6—figure supplement 1*). Estimates of imprinting protection and age-specific risk of medically attended influenza in the birth-cohort-specific VE models are not significantly different from estimates from the best-fitting model fitted to all ages (*Appendix 2—table 1*). Taken together, these results suggest that birth-cohort-specific VE best explains the case distribution in older children and adults, who have likely experienced their first influenza infection, whereas age-specific VE best explains cases in younger children, who have less influenza exposure.

VE differs between birth cohorts that have similar imprinting by subtype (*Figure 6*; *Appendix 2—table 5*). For example, the 1968–1977 and 1988–1997 cohorts have similar probabilities of primary exposure to H1N1 and H3N2, but they differ substantially in their VE to both subtypes (*Figure 6*). The 1988–1997 and 1998–2002 cohorts also have similar probabilities of primary exposure to each subtype and have similar H1N1 VEs, but have significantly different H3N2 VEs (*Figure 6*). Antigenic differences within each subtype might explain this variation.

## Discrepancies partly explained by antigenic evolution

The best-fitting model accurately reproduces the age distributions of vaccinated and unvaccinated cases of each subtype, aggregated across seasons (*Figure 7A*). The only exception is that it underestimates aggregate H1N1 cases in unvaccinated 5–9 year-olds. By examining the differences between predicted and observed cases for each season, we see that this is largely driven by infection during the 2009 H1N1 pandemic (*Figure 7B*). Such a large antigenic change may have negated any protection from previous infection in 5–9 year-olds and made them particularly susceptible to pandemic infection.

The model underestimates cases in unvaccinated individuals who were 30–39 years old and over 50 years old in the 2013–2014 season (*Figure 7B*), as indicated by the many excess cases in these age groups in that season. This is further evidence that subtype-specific imprinting cannot explain all age variation. As mentioned before, this season provided one of the first examples that original antigenic sin could affect protection: middle-aged adults had been targeting a familiar site on the pandemic strain that then mutated, rendering them susceptible. Other age groups were effectively blind to these changes, owing to their different exposure histories (*Linderman et al., 2014*; *Huang et al., 2015*; *Arriola et al., 2014*; *Dávila et al., 2014*; *Petrie et al., 2016*).

## Discussion

The distribution of influenza cases by birth year is consistent with subtype-level imprinting, whereby initial infection with a subtype protects against future medically attended infections by the same subtype. The stronger protective effect observed from primary H1N1 infection compared to primary H3N2 infection may be caused by stronger cross-protective responses to conserved epitopes in the more slowly evolving H1N1 (*Bedford et al., 2015*). This is in line with previous work showing that protection conferred by H1N1 infection lasts longer than protection conferred by H3N2 infection (*Ranjeva et al., 2019*). Another recent study found stronger imprinting protection from primary H1N1 compared to primary H3N2 infection (*Gostic et al., 2019*). Subtype-specific protection observed in seasonal influenza is narrower than the previously reported HA-group-level imprinting protection against avian influenza (*Gostic et al., 2016*), but in both cases, the protection correlates strongly with primary infection rather than any prior exposure.

Examining cases of seasonal influenza over a 20 year period in Arizona, *Gostic et al., 2019* find evidence of imprinting protection not only from HA but also NA, which we do not. We speculate that this discrepancy may be due to increasing vaccination coverage over time in middle-aged

adults. During the period of the Arizona study (1993–1994 through 2014–2015), vaccination coverage in U.S. adults increased most rapidly in this age group (*NHIS, 2009*), which corresponds to the H2N2-imprinted cohorts near the end of the study. Without adjustment for vaccination, the apparently increased protection in the middle aged might resemble N2 imprinting. Accounting for vaccination in the MESA population, including the relatively stable vaccination coverage in each age group over time (*Figure 1—figure supplement 3*), suggests imprinting protection is driven by HA.

In contrast to the clear role of the imprinting subtype in protection against medically attended infection, the model implicates the imprinting strain or other attributes of early exposure history in VE. We expect that people born around the same time were likely exposed to similar strains, not just subtypes, of influenza A early in life, and our results support the idea that biases in immune memory from these early exposures (i.e., original antigenic sin; *Davenport and Hennessy, 1957*; *Francis, 1960*; *Fazekas de St Groth and Webster, 1966*) influence VE. Specifically, we observe that our model is consistent with previous suggestions of birth-cohort-specific VE. The model with birth-cohort-specific VE better estimates cases in vaccinated 50–64 year-olds (born 1953–1967) in the 2015–2016 season than the model with age-specific VE, as indicated by the fewer excess cases predicted in that age group and an improved fit of 1.1 log-likelihood units (*Figure 6—figure supplement 2*; Appendix 1: 'Calculating excess cases'). Reduced VE in this group during the 2015–2016 season has been attributed to the exacerbation of antigenic mismatch by the vaccine in adults whose antibody responses were focused on a non-protective site (*Skowronski et al., 2017b*; *Flannery et al., 2018*). The improved performance of birth-cohort-specific VE relative to age-specific VE suggests other seasons and age groups where original antigenic sin might have influenced VE, such as 20–29 year-olds in the 2007–2008 influenza season.

Although seasonal estimates of VE routinely stratify by age, shifts in VE from one season to the next might thus be easier to interpret in light of infection history (e.g., *Skowronski et al., 2017b*; *Flannery et al., 2018*). The results suggest this effect may be subtle, i.e, influenced by strains' specific identities rather than merely their subtype. Our model cannot distinguish between the possibility that the precise identity of the imprinting strain primarily determines later VE, or if individuals' responses to vaccination are shaped by a particular succession of exposures, which would be common to others in the same birth cohort. Regardless, variation in VE between birth cohorts appears substantial and presents a challenge for vaccination strategies (*Erbelding et al., 2018*).

The use of different influenza vaccines in MESA during this period is unlikely to affect the results. Most people enrolled in the study received the standard-dose inactivated influenza vaccine (IIV-SD) (*Figure 1—figure supplement 7*). However, between 9–26% of vaccinated children <18 years old received the live attenuated influenza vaccine (LAIV) between the 2008–2009 and 2015–2016 seasons (*Figure 1—figure supplement 7B*). A separate study of LAIV VE in the United States found that LAIV and IIV-SD recipients who were repeat vaccinees (as most children were) had similar VE, and thus we do not expect that LAIV receipt should affect VE estimates (*McLean et al., 2018*). Similarly, 1–15% of adults ≥65 years old received the high-dose inactivated influenza vaccine (IIV-HD) between 2009–2010 and 2017–2018 (*Figure 1—figure supplement 7C*). This vaccine is 20% more effective than IIV-SD (*Lee et al., 2018*). Therefore, the changing ratio of IIV-HD to IIV-SD recipients over time might bias results toward cohort-specific VE in models that include people ≥65 years old. However, when we fitted to cases between 15–64 years old, we found that cohort-specific VE still performed best. Thus, we conclude that changes in IIV-HD coverage do not substantially influence results.

Potential methodological biases and the vaccination history of the study population nonetheless suggest caution in interpreting VE estimates. Selection and misclassification biases can arise when using influenza test-negative controls to control for differences in healthcare-seeking behavior (*Lewnard et al., 2018*; *Sullivan et al., 2016*). Because we also use test-negative controls to set our null expectation for the distribution of cases among birth cohorts, our VE estimates are subject to these biases as well. Moreover, since 45% of the study population is vaccinated, and most participants are frequent vaccinees (*Figure 1—figure supplement 6*), we are limited in our ability to generalize the VE results to populations with much lower vaccination coverage and/or a shorter history of vaccination. Frequent vaccination has been associated with reduced VE (*McLean et al., 2014*; *Saito et al., 2018*; *Skowronski et al., 2016*). Therefore, the model may underestimate VE in less vaccinated populations. Underestimation of VE could also occur if unvaccinated people are protected by vaccination in the preceding season. Inference might also be distorted if vaccination has

large indirect effects, which our model does not consider. Finally, our analysis is worth repeating in a larger population to reduce stochastic influences. We observed an unusually high H1N1 VE in the 1998–2002 birth cohort. Because we restricted cases in this analysis to people ≥15 years old, this VE estimate included data from only the 2013–2014 and 2015–2016 influenza seasons. No H1N1 cases among vaccinated or unvaccinated individuals were observed in this birth cohort in those seasons, which led to the high VE. This might have been due to particular epidemic dynamics in MESA.

Incorporating differences in susceptibility based on early exposures might improve methods to forecast influenza seasons. The analysis of the relative risk of infection during the first half of each season shows more variation in the susceptible age groups from season to season than previously estimated (*Worby et al., 2015*). While the smaller sample sizes in MESA introduce uncertainty, the correlation between the relative risk and total fraction of cases indicates that the age groups driving epidemics indeed change from season to season. Because the contact structure of the population is probably constant over influenza seasons, variation in the driving age group may be determined by fluctuating susceptibility, which is partly determined by early infections. Therefore, incorporating information on early exposure history into epidemic models may allow for more accurate identification of at-risk populations and fine-scale epidemic timing.

While the rate of antigenic evolution affects the rate at which different populations become susceptible to infection, we propose that the heterogeneity in susceptibility observed here may also drive antigenic evolution. Heterogeneity in susceptibility implies that influenza viruses face different selective pressures in groups with different exposure histories (*Cobey and Koelle, 2008*; *Nakajima et al., 2000*). Recent research consistent with this hypothesis has shown that sera isolated from different individuals can select for distinct escape mutants (*Lee et al., 2019*). More careful study of how immune memory to influenza evolves from infection and vaccination might improve understanding of influenza's evolution.

## Code and data availability

The code and data used to perform the analyses for this project are available at https://github.com/cobeylab/FluAImprinting (*Arevalo et al., 2019*; copy archived at https://github.com/elifesciences-publications/FluAImprinting).

## Acknowledgements

We thank Jennifer King and Carla Rottscheit for their assistance in providing the data for this study and Rohan Dandavati for compiling historical data on subtype frequencies and ILI. We thank Marcos Vieira and Kangchon Kim for their assistance in calculating imprinting probabilities. This work was completed with computational resources provided by the University of Chicago's Research Computing Center. Funding for this project was provided by the National Institutes of Health (NIH), Department of Health and Human Services, under grant DP2AI117921 (to SC), CEIRS Contract No. HHSN272201400005C (to SC), and NRSA Fellowship F32AI145177-01 (to PA). HQM receives research support from Seqirus unrelated to this work. The funders had no role in study design, data collection and analysis, decision to publish, or preparation of the manuscript.

## Additional information

### Competing interests

Huong Q McLean: has received funding from Seqirus, unrelated to this work. The author has no other competing interests to declare. The other authors declare that no competing interests exist.

### Funding

| Funder | Grant reference number | Author |
| --- | --- | --- |
| National Institutes of Health | DP2AI117921 | Sarah Cobey |
| National Institutes of Health | F32AI145177-01 | Philip Arevalo |
| National Institutes of Health | HHSN272201400005C | Sarah Cobey |

The funders had no role in study design, data collection and interpretation, or the decision to submit the work for publication.

## Author contributions
Philip Arevalo, Conceptualization, Software, Formal analysis, Visualization, Methodology; Huong Q McLean, Conceptualization, Data curation, Funding acquisition, Project administration; Edward A Belongia, Conceptualization, Resources, Funding acquisition, Project administration; Sarah Cobey, Conceptualization, Resources, Formal analysis, Supervision, Funding acquisition, Methodology, Project administration

## Author ORCIDs
Philip Arevalo (iD) https://orcid.org/0000-0003-1237-2314

## Ethics
Human subjects: Study procedures for the vaccine effectiveness study was approved by the IRB at the Marshfield Clinic Research Institute. Informed consent was obtained from all participants at the time of enrollment into the vaccine effectiveness study. This analysis was subsequently approved by the Marshfield Clinic Research Institute IRB with a waiver of informed consent. The analysis of data was approved by the University of Chicago IRB under protocol number IRB17-1134-CR001.

## Decision letter and Author response
Decision letter https://doi.org/10.7554/eLife.50060.sa1
Author response https://doi.org/10.7554/eLife.50060.sa2

# Additional files
## Supplementary files
• Transparent reporting form

## Data availability
Code and data for calculation of imprinting probabilities, vaccination coverage, and model fitting are available on GitHub at https://github.com/cobeylab/FluAImprinting (copy archived at https://github.com/elifesciences-publications/FluAImprinting).

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

## Appendix 1

### Vaccination coverage

Seasonal influenza vaccination coverage for MESA was collected by age in the 2007–2008 through 2017–2018 seasons using a regional immunization registry (*Irving et al., 2009*). Monovalent vaccination coverage for the 2009–2010 season was obtained by directly measuring monovalent vaccination coverage in enrolled individuals and fitting a smoothing spline to the data (*Figure 1—figure supplement 3*). We also calculated the fraction of people who received different vaccination formulations, and found that most people received IIV-SD (*Figure 1—figure supplement 7*).

### Correlation of relative risk and fraction of cases

To assess whether an age group's relative risk correlates with the fraction of cases of that age group in the same season, we performed a rank correlation analysis. For each season, we ranked each age group based on its relative risk and the fraction of cases within that age group. If age groups were tied in either relative risk or fraction of cases, we assigned them the average rank they spanned. We then calculated the Pearson's correlation coefficient for these two rankings. A positive correlation indicates that an age group with a large relative risk compared to other age groups will also make up a large proportion of cases compared to other age groups.

### Seasonal intensity

We defined the intensity of an influenza season as the product of the mean fraction of patients with influenza-like illness (ILI) and the percentage of specimens testing positive for influenza A that season,

$$I_t = \frac{\text{ILI}_t F_t}{N_t},$$ (28)

where $\text{ILI}_t$ is the mean fraction of all patients with ILI in season $t$ adjusted for differences in state population size (*CDC, 2018*), $F_t$ is the number of respiratory specimens testing positive for influenza A in season $t$, and $N_t$ is the total number of respiratory specimens tested in season $t$. For seasons 1997–1998 through 2017–2018, these data were obtained from the U.S. Outpatient Influenza-like Illness Surveillance Network (ILINet) and the World Health Organization/National Respiratory and Enteric Virus Surveillance System (WHO/NREVSS) Collaborating Labs (*CDC, 2018*). For seasons 1976–1977 through 1996–1997 when seasonal ILI data were not available, we assumed that the mean ILI was equal to the mean of mean ILI for seasons 1997–1998 through 2017–2018. We obtained data on $F_t$ and $N_t$ for these seasons from *Thompson et al., 2003*. We then normalized the intensity of each season by dividing $I_t$ by the mean of $I_t$ from the 1976–1977 through 2017–2018 seasons. For all seasons before 1976–1977, for which no seasonal intensity data were available, we assumed that the intensity of influenza A equalled the mean intensity of seasons 1976–1977 through 2017–2018.

### Fraction of season experienced

We defined the fraction of a given influenza season $f_{w,t}$ occurring in week $w$ of season $t$ as

$$f_{w,t} = \frac{\text{ILI}_{w,t} F_{w,t}}{N_{w,t} \sum_{w'=w_0}^{w_f} \frac{\text{ILI}_{w',t} F_{w',t}}{N_{w',t}}},$$ (29)

where $\text{ILI}_{w,t}$ is the weighted fraction of all patients with ILI in week $w$ of season $t$, $F_{w,t}$ is the number of respiratory specimens testing positive for influenza A in week $w$ of season $t$, and $N_{w,t}$ is the number of specimens tested in week $w$ of season $t$. $\sum_{w'=w_0}^{w_f} \frac{\text{ILI}_{w',t} F_{w',t}}{N_{w',t}}$ is the product of ILI and the fraction of positive influenza A specimens summed over all weeks of the influenza season $t$, where $w_0$ is the first week of the season and $w_f$ is the final week of the season. We defined the start of the influenza season as week 40 of the calendar year, which usually falls at the beginning of October. For seasons

before 1997–1998, where weekly data is unavailable, we assumed that the fraction of the influenza season experienced in week $w$ was

$$f_{w,t} = \bar{f}_{w,t},$$ (30)

where $\bar{f}_{w,t}$ is the mean fraction of the influenza season experienced at week $w$ for all seasons after 1997–1998.

We used $f_{w,t}$ to calculate the fraction of an influenza season experienced by an individual born in year $y$. We assumed that people born in year $y$ were born uniformly throughout the year. We also assumed that due to maternal immunity, infants did not experience immunizing exposure to influenza until they were at least 180 days old. Let $p_{y,w,t}$ be the proportion of individuals born in year $y$ that are over 180 days old in week $w$ of season $t$ and $\gamma_{y,t}$ be the fraction of individuals born in year $y$ exposed to influenza season $t$. Then

$$\gamma_{y,t} = \sum_{w=w_0}^{w_f} f_{w,t} p_{y,w,t}.$$ (31)

## Calculating the fraction unexposed

When calculating imprinting probabilities, we used an iterative approach to calculate $U_{y,t}$ the fraction of people in birth cohort $y$ who were unexposed at the start of season $t$. First, we assumed that in the first year of life (i.e., when $t = y$), the entire population was unexposed. For seasons where $t > y$, the fraction unexposed depends on the fraction unexposed at the start of the previous season ($U_{y,t-1}$) and the attack rate in the previous season ($a_{y,t-1}$). Thus,

$$U_{y,t} = \begin{cases} 1 & t=y \\ U_{y,t-1}(1 - a_{y,t-1}) & t > y \end{cases}$$ (32)

## Birth year distribution of the study population

In order to convert the demographic age distribution to a birth year distribution, we assumed that people were born uniformly throughout the year. We defined a breakpoint date prior to the start of the enrollment period based on when the the 6 month-old age limit cutoff was set (e.g., if the breakpoint date was Ocotober 1, then infants had to be 6 months old by that date to be eligible for enrollment). We used this date to calculate the fraction of people of age $a$ in season $t$ who were born in year $t - y$ ($d^1_{a,t,y}$) or year $t - y - 1$ ($d^2_{a,t,y}$). A fraction $d^1_{a,t,y}$ of the total population of age $a$ in season $t$ was assigned to birth year $t - y$ and $d^2_{a,t,y}$ to $t - y - 1$. Breakpoint dates ranged from September one through January one with the exception of the pandemic season which had a breakpoint date of May 1, 2009. The start of the enrollment period ranged from December to January with the exception of the 2009 pandemic season, when enrollment began in May 2009.

## Fraction of birth cohort with specific age

When converting an age-specific parameter to a birth-cohort-specific parameter as in Materials and Methods 'Age-specific factors', we considered that each birth cohort had two possible ages ($a1$ and $a2$) in a given season $t$. We assumed that people were born uniformly throughout the year and used the same breakpoint dates described above in 'Birth year distribution of the study population.' Then, $f(a1, t, y)$, the fraction of people born in year $y$ who were age $a1$ in season $t$, is the fraction of people born in year $y$ who were born on a date prior to the breakpoint date for season $t$. Finally, $f(a2, t, y)$, the fraction of people born in year $y$ who were age $a2$ in season $t$, is 1 - $f(a1, t, y)$.

## Age-specific rates of approachment, enrollment, and nursing home residence

The relative rates at which different age groups were approached for study enrollment (the approachment rate, $x_{\text{approach}}$) varied between seasons. Similarly, the relative rates at which different age groups enrolled in the study after being approached (the enrollment rate, $x_{\text{enroll}}$) also varied between seasons. Enrollment rates also varied between vaccinated and unvaccinated individuals.

We defined the approachment rate of an age group $g$ in season $t$ as

$$x_{\text{approach},t,g} = \frac{N_{\text{approached},t,g}}{N_{\text{MAARI},t,g}}, \tag{33}$$

where $N_{\text{approached},t,g}$ is the number of people in age group $g$ during season $t$ who were approached for enrollment, and $N_{\text{MAARI},t,g}$ is the total number of people in the MESA cohort who presented with MAARI regardless of whether they were approached for enrollment.

We defined the enrollment rate of age group $g$ in season $t$ with vaccination status $v$ as

$$x_{\text{enroll},t,g,v} = \frac{N_{\text{enrolled},t,g,v}}{N_{\text{approached},t,g,v}} \tag{34}$$

where $N_{\text{enrolled},t,g,v}$ is the number of people in age group $g$ with vaccination status $v$ who enrolled in the study in season $t$, and $N_{\text{approached},t,g,v}$ is the number of people in age group $g$ with vaccination status $v$ who were approached for enrollment in season $t$. Due to differences in data collection for the 2007–2008 and 2008–2009 seasons, complete vaccination records for eligible unenrolled individuals were not available, so we assumed that the enrollment rates by age group and vaccination status in those seasons were equal to the mean enrollment rate for each age group and vaccination status across all other seasons.

We normalized $x_{\text{approach},t,g}$ by the value of $x_{\text{approach},t,g}$ for the reference age group (i.e., 20–29 year-olds) in each season. Similarly, we normalized $x_{\text{enroll},,t,g,v}$ to the value of $x_{\text{enroll},,t,g,v}$ for unvaccinated members of the reference age group for each season. This yielded the relative approachment and enrollment rates $x'_{\text{approach},t,g}$ and $x'_{\text{enroll},t,g,v}$. We converted both $x'_{\text{approach},t,g}$ and $x'_{\text{enroll},t,g,v}$ to birth-year specific covariates (i.e. covariates by $y$ instead of $g$) using the same procedure described in Materials and Methods: 'Age-specific factors' (*Equation 9*).

Finally, the study did not enroll residents of skilled nursing facilities with dedicated medical staff. To account for this, we estimated the proportion of the population in nursing facilities within the study area. We obtained the total number of beds in nursing facilities within MESA in 2018 from the Wisconsin Department of Health Services (*WDHS, 2018*). We assumed that the total number of beds did not change between 2007–2008 and 2017–2018. We also used data from the Centers for Medicare and Medicaid Services (*CMS, 2015*) to calculate the percent of beds occupied in Wisconsin nursing facilities by age for 2011 through 2014 and the fraction of people in a nursing facility by age group. We used a smoothing spline to obtain the fraction of people of a given age in a nursing facility. For seasons before 2010–2011 and after 2013–2014, we assumed that the fraction of people of a given age in a nursing facility was the average value for 2011–2014. Given the total population of the study area by age and season, we calculated the fraction of people in a given age $a$ and season $t$ who are in nursing facilities ($k_{t,a}$). We converted this to a covariate by birth year ($k_{t,y}$) using the same procedure described in Materials and Methods: 'Age-specific factors' (*Equation 9*).

## Evaluation of predictive power

To evaluate the predictive power of each model, we performed leave-one-out cross-validation. We excluded data from each season and fitted our models to the remaining seasons. Because our goal was to evaluate how well our models predict seasonal epidemics, we excluded the 2009 pandemic season from all cross-validation analyses. We also did not test seasonal VE models with cross-validation since estimation of seasonal VE requires data from the excluded season.

Let $n_{s,t,y,v}$ be the number of observed cases of subtype $s$ in season $t$ among people born in year $y$ with vaccination status $v$. Let $p_{M,s,t^-,y,v}^{t^-}$ be the multinomial probability of a case of subtype $s$ in season $t^-$ among people born in year $y$ with vaccination status $v$ under model $M$ fitted to all seasons except $t^-$. Let $N_{s,t^-}$ be the total number of cases of subtype $s$ in season $t^-$. Then, the predicted number of cases of subtype $s$ in season $t^-$ among people born in year $y$ with vaccination status $v$ under model $M$ fitted to all seasons except $t^-$ is

$$p_{M,s,t^-,y,v}^{t^-} N_{s,t^-}. \tag{35}$$

The sum squared prediction error for model $M$ in season $t^-$ is given by

$$\text{SSE}_{M,t^-} = \sum_{y=1918}^{y_{\max,t}} (n_{s,t^-,y,\text{unvac.}} - p^{t^-}_{M,s,t^-,y,\text{unvac.}} N_{s,t^-})^2 + \sum_{y=1918}^{y_{\max,t}} (n_{s,t,y,\text{vac.}} - p^{t^-}_{M,s,t^-,y,\text{vac.}} N_{s,t^-})^2, \tag{36}$$

where $y_{\max,t}$ is the maximum possible birth year in season $t$.

We evaluated each model $M$ by its mean-squared prediction error across all excluded seasons $t^-$. Let $T^-$ be the set of all seasons left out and $X$ be the size of $T^-$. Then the mean-squared prediction error for model $M$ is

$$\text{MSE}_M = \frac{\sum_{t^- \in T^-} \text{SSE}_{M,t^-}}{X}. \tag{37}$$

## Sensitivity to uncertainty in ILI and the frequency of influenza A

Because of the lack of ILI data prior to the 1997–1998 season and the lack of data on the frequency of influenza A prior to the 1976–1977 season, we used simulated datasets to test the robustness of our results. We randomly assigned ILI values from the 1997–1998 through 2017–2018 seasons to every season which did not have a measured ILI value. Similarly, we randomly assigned values of the frequency of influenza A from the 1976–1977 through 2017–2018 seasons to every season which did not have a measured value for the frequency of influenza A. We created 10000 simulated datasets using this procedure and recalculated imprinting probabilities for each dataset (*Figure 3—figure supplement 2*). In the period of H1N1 and H3N2 co-circulation, the maximum H1N1 imprinting probability for a particular birth cohort corresponds to the minimum H3N2 imprinting probability for that cohort and vice-versa. Therefore, to generate datasets representing the upper and lower bounds of imprinting probabilities, we assigned imprinting probabilities from the simulation with either the lowest or highest H1N1 imprinting probability to each birth cohort in each season. We then fitted our models to these two datasets and evaluated model fit using cAIC.

## Sensitivity to age groups

To test whether our models were sensitive to our choice of age groups, we fit revised versions of all our models with different age groups:

- 0–4 years, 5–17 years, 18–49 years, 50–64 years, and ≥65 years
- 0–4 years, 5–17 years, 18–64 years, and ≥65 years

These models with alternate age groupings were fitted to case data to determine whether our findings on the strength of protection from initial H1N1 and H3N2 infection were altered compared to fits using the higher-resolution age grouping described above (*Appendix 2—table 4*).

## Sensitivity to sampling effort

Sampling effort was not even across seasons, and analysis of the number of influenza cases per sampling day suggested that a significant number of cases may have been missed at the beginning or end of a specific seasons (*Figure 5—figure supplement 3*). As our analysis of relative risk indicates, different age groups are more susceptible during different points in the influenza season, and therefore missing data from the beginning or end of a season could introduce bias in the observed age distribution of cases.

To adjust for this, we simulated cases for seasons which did not have sufficient sampling of the start or end of the epidemic period. We considered a season sufficiently sampled if the sampling period spanned the start and end of the epidemic. We expect that the start and end of the epidemic have few cases per sampling day, and we therefore defined sufficiently sampled seasons as seasons where

- the number of cases per sampling day in the first week of the enrollment period is < 1 and
- the number of cases per sampling day in the last week of the enrollment period is < 1.

To extrapolate the start of a season, we linearly regressed the number of cases of the dominant subtype per sampling day for each week of the first half of the season and identified the week of the season where the number of cases per sampling day fell below 1 ($t_0$). For each week from $t_0$ to the

first week of the enrollment period, we used the regression of cases per sampling day to calculate the number of cases we expected to see in each week. Summing these yields the total number of unsampled cases at the beginning of the season. We used a similar approach to extrapolate the number of unsampled cases at the end of a season by instead regressing cases per sampling day for each week of the latter half of the season. We did not extrapolate cases for the 2010–2011 season for this analysis since the observed number of cases per sampling day did not follow a typical epidemic curve.

We stochastically assigned a birth year and vaccination status to these cases according to a multinomial distribution. The success probabilities of this distribution were set using the age distribution of cases of the dominant subtype from the first two weeks of the enrollment period (if extrapolating the beginning of a season) or the last two weeks of the enrollment period (if extrapolating the end of a season). Specifically, we calculated the distribution of observed cases in the first or last two weeks of the enrollment period among nine age groups (Materials and Methods: 'Age-specific factors') with their associated vaccination status. We then assumed that cases were uniformly distributed among all birth years contained in an age group. This yielded a set of probabilities describing the probability of infection given birth year and vaccination status in a specific season.

We sampled from these multinomial distributions 1000 times to obtain augmented datasets that combined observed and extrapolated cases. For each replicate simulation, we calculated the age distribution of cases for the entire season as well as the relative risk of each age group in the first versus the latter half of the season (*Figure 2—figure supplement 2B*). We also fitted the best-fitting model to 100 of these datasets (excluding the 2010–2011 season) and recorded the estimated imprinting strength for both H1N1 and H3N2 for each fit (*Figure 5—figure supplement 4*).

## Calculating excess cases

We defined excess cases for a given birth cohort or age group as the number of observed cases for that birth cohort or age group minus the number of predicted cases for that age group. Predictions were obtained by multiplying the multinomial probabilities produced by the model by the total number of cases of the dominant subtype in each season. A 95% prediction interval was obtained by simulating 10000 datasets using the multinomial probabilities from a specific model (*Figure 6—figure supplement 2*, *Figure 7*).

To test whether recent infection might be confounding our estimates, we calculated the correlation between excess cases in each birth cohort in each season with excess cases of the same birth cohort in the next season with the same dominant subtype (*Figure 5—figure supplement 5*).

# Appendix 2

## Supplementary tables and figures

**Appendix 2—table 1.** Estimates of parameters shared by the age-specific VE and birth-cohort-specific VE models.

| | Model with age-specific VE, age ≥6 months (MLE, 95% CI) | Model with age-specific VE, age ≥15 years (MLE, 95% CI) | Model with age-specific VE, age < 65 years (MLE, 95% CI) | Model with age-specific VE, age 15–64 years (MLE, 95% CI) | Model with birth-cohort-specific VE, age ≥15 years (MLE, 95% CI) |
|---|---|---|---|---|---|
| Imprinting protection (%) | | | | | |
| H1 | 66 (53, 77) | 48 (25, 66) | 64 (47, 77) | 43 (11, 66) | 49 (24, 67) |
| H3 | 33 (17, 46) | 41 (20, 56) | 34 (18, 47) | 36 (13, 52) | 41 (20, 56) |
| N2 | 0 (0, 7) | 0 (0, 11) | 0 (0, 8) | 0 (0, 10) | 0 (0, 11) |
| Age-specific risk of medically attended influenza A infection | | | | | |
| 0–4 years | 3.0 (2.5, 3.6) | N.A. | 3.0 (2.5, 3.6) | N.A. | N.A. |
| 5–9 years | 2.6 (2.2, 3.0) | N.A. | 2.5 (2.2, 3.0) | N.A. | N.A. |
| 10–14 years | 1.7 (1.4, 2.0) | N.A. | 1.7 (1.4, 2.0) | N.A. | N.A. |
| 15–19 years | 1.2 (1.0, 1.5) | 1.2 (1.0, 1.5) | 1.2 (1.0, 1.5) | 1.2 (1.0, 1.5) | 1.2 (1.0, 1.5) |
| 30–39 years | 1.1 (0.9, 1.3) | 1.1 (0.9, 1.3) | 1.1 (0.9, 1.3) | 1.1 (0.9, 1.3) | 1.1 (0.9, 1.3) |
| 40–49 years | 0.9 (0.7, 1.1) | 0.9 (0.8, 1.1) | 0.9 (0.7, 1.1) | 0.9 (0.8, 1.1) | 0.9 (0.8, 1.1) |
| 50–64 years | 1.0 (0.8, 1.3) | 1.0 (0.8, 1.2) | 1.0 (0.8, 1.3) | 1.0 (0.8, 1.2) | 0.9 (0.7, 1.1) |
| 65+ years | 1.6 (1.2, 2.1) | 1.4 (1.0, 1.9) | N.A | N.A. | 1.5 (1.1, 1.9) |

**Appendix 2—table 2.** Estimates of age-specific VE parameters in models fitted to different age groups.

| | Model with age-specific VE, age ≥6 months (MLE, 95% CI) | Model with age-specific VE, age ≥15 years (MLE, 95% CI) | Model with age-specific VE, age < 65 years (MLE, 95% CI) | Model with age-specific VE, age 15–64 years (MLE, 95% CI) |
|---|---|---|---|---|
| Age-specific VE against H1N1 (%) | | | | |
| 0–4 years | 69 (56, 84) | N.A. | 68 (55, 83) | N.A. |
| 5–9 years | 26 (0, 48) | N.A. | 24 (0, 47) | N.A. |
| 10–14 years | 92 (80, 96) | N.A. | 92 (80, 96) | N.A. |
| 15–19 years | 86 (62, 95) | 89 (66, 97) | 86 (61, 95) | 89 (65, 97) |
| 20–29 years | 84 (65, 91) | 86 (69, 91) | 83 (63, 90) | 85 (67, 91) |
| 30–39 years | 8 (0, 37) | 22 (0, 47) | 5 (0, 35) | 19 (0, 45) |
| 40–49 years | 18 (0, 45) | 28 (0, 47) | 14 (0, 42) | 24 (0, 49) |
| 50–64 years | 32 (7, 51) | 39 (16, 56) | 28 (2, 48) | 37 (14, 55) |
| 65+ years | 50 (16, 71) | 64 (39, 83) | N.A. | N.A. |
| Age-specific VE against H3N2 (%) | | | | |
| 0–4 years | 58 (48, 67) | N.A. | 58 (48, 67) | N.A. |
| 5–9 years | 45 (31, 58) | N.A. | 45 (30, 57) | N.A. |
| 10–14 years | 23 (0, 41) | N.A. | 22 (0, 41) | N.A. |
| 15–19 years | 31 (3, 53) | 33 (4, 55) | 30 (2, 53) | 32 (1, 54) |
| 20–29 years | 34 (11, 51) | 37 (15, 53) | 33 (11, 51) | 36 (14, 53) |
| 30–39 years | 10 (0, 31) | 15 (0, 35) | 9 (0, 30) | 12 (0, 33) |
| 40–49 years | 36 (15, 52) | 42 (24, 57) | 36 (15, 52) | 42 (23, 57) |
| 50–64 years | 47 (35, 56) | 49 (37, 58) | 47 (35, 57) | 48 (36, 58) |
| 65+ years | 41 (24, 54) | 38 (20, 52) | N.A. | N.A. |

**Appendix 2—table 3.** Estimates of imprinting protection fitted to datasets representing upper and lower bounds of imprinting probabilities.

| Dataset | Best-fitting model | H1 imprinting protection (%, 95% CI) | H3 imprinting protection (%, 95% CI) |
|---|---|---|---|
| Lower bound | Demography, age, HA imprinting, age-specific VE | 72 (57, 84) | 32 (17, 44) |
| Upper bound | Demography, age, HA imprinting, age-specific VE | 61 (48, 72) | 37 (20, 51) |

**Appendix 2—table 4.** Estimates of imprinting protection for models with different age groups.

| Age groups (years) | Best-fitting model | H1 imprinting protection (%, 95% CI) | H3 imprinting protection (%, 95% CI) |
|---|---|---|---|
| 0–4, 5–17, 18–64, 65+ | Demography, age, HA imprinting, age-specific VE | 56 (40, 68) | 36 (25, 46) |
| 0–8, 9–17, 18–49, 50–64, 65+ | Demography, age, HA imprinting, age-specific VE | 62 (47, 74) | 35 (21, 48) |

**Appendix 2—table 5.** Estimates for VE from model with birth-cohort-specific VE fitted to people ≥15 years old.

| Birth cohort | H1N1 VE (%, MLE, 95% CI) | H3N2 VE (%, MLE, 95% CI) |
|---|---|---|
| 1998–2002 | 100 (22, 100) | 0 (0, 36) |
| 1988–1997 | 89 (74, 93) | 62 (45, 76) |
| 1978–1987 | 59 (35, 76) | 17 (0, 35) |
| 1968–1977 | 23 (0, 47) | 25 (2, 44) |
| 1953–1967 | 28 (4, 46) | 43 (32, 53) |
| 1918–1952 | 61 (38, 76) | 45 (32, 55) |

