## [Decision Letter]

**Acceptance summary:**

Rapid evolution of influenza A presents the immune system with a moving target and allows the virus to infect the same human host multiple times. This is because changes in the virus mean that acquired immunity to a strain causing one infection will not, in general, be able to prevent subsequent infections with different strains. The first influenza infection a person experiences may have a particularly strong influence on how the immune system responds to subsequent influenza infections. In this study, the effects of these early influenza exposures (determined indirectly by birth year) are examined with a thorough and elegant analysis of high quality influenza data. The findings provide strong evidence that the first infection does indeed have an important influence, explaining variation in the age distribution of influenza cases in different influenza seasons and variation in vaccine effectiveness. The evidence that vaccine effectiveness varies with birth cohort and not just with age, in particular, has important implications for how we think about vaccine effectiveness.

**Decision letter after peer review:**

Thank you for submitting your article "Earliest infections predict the age distribution of seasonal influenza A cases" for consideration by *eLife*. Your article has been reviewed by four peer reviewers, including Ben S Cooper as the Reviewing Editor and Reviewer #2, and the evaluation has been overseen by Neil Ferguson as the Senior Editor. The following individual involved in review of your submission has agreed to reveal their identity: Marc Baguelin (Reviewer #3).

The reviewers have discussed the reviews with one another and the Reviewing Editor has drafted this decision to help you prepare a revised submission.

Summary:

This manuscript presents an analysis of 10 years of clinical testing of respiratory illness cases for influenza from a single population in the US. Data include PCR-confirmed influenza cases including type and sub-type, age and vaccine history. The analysis aims to quantify the effects of earliest influenza infections in childhood on subsequent risk of clinical infection with different subtypes, a question of considerable scientific interest. The analysis provides evidence that susceptibility to different subtypes is determined by the subtype of the first infection (as proxied by the birth cohort) and that vaccine responses are driven by precise infection history.

Essential revisions:

1) Fairly substantial re-organisation of the paper is needed in order to improve readability. In particular, although the Results and Discussion are separate sections, the Results sections tend to stray into discussion, and we advise you to change this. Furthermore, this manuscript would profit from putting the Materials and methods before the Results. Although it is normal to structure *eLife* articles with the Materials and methods section following the Discussion, we can accommodate different organisation if the editors feel the paper requires this for readability.

2) In some places more precise terminology is needed and the almost interchangeable use of infection and case is potentially confusing. These data are all obtained from people seeking care and the primary results are about subtype differences. At the very least, the language describing the results has to be very tightly constrained to cases with any discussion of possible implications for infections reserved for the discussion. Although there is a paucity of good data describing the rate at which H1 and H3 result in clinical cases, there is no suggestion that this is a uniform process. It likely varies from year to year and from age group to age group.

3) The formal description of the models was in some places ambiguous or confusing and this section needs careful attention.

4) We strongly suggest that the results should be about cases as a function of birth cohort (as these are the data available) and then the idea that birth cohort may be a proxy for infection history should be reserved for the discussion. Even in older people, recent infection history will vary among people the same age and will likely to be a dominant factor in both risk of infection and risk of clinical disease. Especially, given other publications from the group looking at epitope-level histories, mapping infection history so strongly onto birth cohort seems unusual.

5) The results are likely to be of wide interest because of the duration of the study and the consistency of the study design. However, the population from which they come is not representative. Levels of vaccination in the US are extremely high and, given how vaccination efficacy is measured by reduced rates of clinical disease rather than by reduced rates of infection, and these data are from clinical cases, caution is needed when generalizing these results. Even though this is mentioned in the discussion, the characteristics of the population are not given sufficient prominence in the Abstract and Discussion.

6) Please provide a visualisation of the raw data showing how many tests were done in each year, and what was their outcome, stratified by reported vaccination status. How did the control population compare?

7) Throughout the manuscript the authors dichotomise results into significant/non-significant based on whether p-values are below or above the arbitrary threshold of 0.05 (and in some cases it is only reported whether p-values are above or below, the actual values are not given). While this is common practice, it goes against mainstream statistical thinking which advises against the use of such "bright lines" when reporting or interpreting results (see the recent consensus statement on p-values from the American Statistical Association which "is intended to steer research into a post p<0.05 era." https://www.amstat.org/asa/files/pdfs/P-ValueStatement.pdf). While p-values clearly have a role (though the authors should bear in mind that with enough data, an arbitrarily small difference of no clinical significance can have an arbitrarily small p-value) dichotomising results into significant /non-significant is rarely helpful. We strongly encourage the authors to consider whether this approach is justified in light of the ASA statement and to consider revising the manuscript accordingly.

8) While not an essential revision, out-of-sample validation of the models (rather than just model comparison) would greatly strengthen the conclusions.

---

## [Author Response]

Essential revisions:1) Fairly substantial re-organisation of the paper is needed in order to improve readability. In particular, although the Results and Discussion are separate sections, the Results sections tend to stray into discussion, and we advise you to change this. Furthermore, this manuscript would profit from putting the Materials and methods before the Results. Although it is normal to structure eLife articles with the Materials and methods section following the Discussion, we can accommodate different organisation if the editors feel the paper requires this for readability.

We have reorganized the paper. The Materials and methods section is now before the Results and we moved less central methods to a Supplementary Methods section to improve readability. We moved the “Modeling approach” section to the Materials and methods directly before the section now titled “Mathematical expressions for model components” to clarify the logic of our approach.

We also increased the distinction between the Results and Discussion sections. For instance, we moved interpretation of the results in the subsection titled “VE varies by birth cohort in older children and adults” to the Discussion. More description is in the responses to Major Point 4.

2) In some places more precise terminology is needed and the almost interchangeable use of infection and case is potentially confusing. These data are all obtained from people seeking care and the primary results are about subtype differences. At the very least, the language describing the results has to be very tightly constrained to cases with any discussion of possible implications for infections reserved for the discussion. Although there is a paucity of good data describing the rate at which H1 and H3 result in clinical cases, there is no suggestion that this is a uniform process. It likely varies from year to year and from age group to age group.

This is an excellent point. We now refer to the risk of “medically attended infections” instead of “infections” where appropriate.

3) The formal description of the models was in some places ambiguous or confusing and this section needs careful attention.

We have made the changes suggested. In brief, we added equation numbers and more detail to the sections on calculating the attack rate, imprinting probabilities, and model likelihood.

4) We strongly suggest that the results should be about cases as a function of birth cohort (as these are the data available) and then the idea that birth cohort may be a proxy for infection history should be reserved for the discussion. Even in older people, recent infection history will vary among people the same age and will likely to be a dominant factor in both risk of infection and risk of clinical disease. Especially, given other publications from the group looking at epitope-level histories, mapping infection history so strongly onto birth cohort seems unusual.

We have shifted most interpretation to the Discussion:

“We hypothesized that VE in these ages varies with early exposure history, which correlates with birth year, rather than age.”

“VE differs between birth cohorts that have similar imprinting by subtype.”

Paragraph beginning “Our results support the idea that biases in immune memory from early exposures (i.e., original antigenic sin) influence VE…” has been moved from Results to Discussion.

5) The results are likely to be of wide interest because of the duration of the study and the consistency of the study design. However, the population from which they come is not representative. Levels of vaccination in the US are extremely high and, given how vaccination efficacy is measured by reduced rates of clinical disease rather than by reduced rates of infection, and these data are from clinical cases, caution is needed when generalizing these results. Even though this is mentioned in the discussion, the characteristics of the population are not given sufficient prominence in the Abstract and Discussion.

We agree that the 45% vaccination coverage of the Marshfield population, although similar to national coverage, limits our ability to generalize some results to much less vaccinated populations. We have revised the Abstract:

“Seasonal variation in the age distribution of influenza A cases suggests that factors other than age shape susceptibility to medically attended infection. We ask whether these differences can be partly explained by protection conferred by childhood influenza infection, which has lasting impacts on immune responses to influenza and protection against new influenza A subtypes (phenomena known as original antigenic sin and immune imprinting). Fitting a statistical model to data from studies of influenza vaccine effectiveness (VE), we find that primary infection appears to reduce the risk of medically attended infection with that subtype throughout life. This effect is stronger for H1N1 compared to H3N2. Additionally, we find evidence that VE varies with both age and birth year, suggesting that VE is sensitive to early exposures. Our findings may improve estimates of age-specific risk and VE in similarly vaccinated populations and thus improve forecasting and vaccination strategies to combat seasonal influenza.”

We have also added the following line to the Discussion:

“Moreover, since 45% of the study population is vaccinated, and most participants are frequent vaccinees (Figure 1—figure supplement 6), we are limited in our ability to generalize the VE results to populations with much lower vaccination coverage and/or a shorter history of vaccination. Frequent vaccination has been associated with reduced VE (McLean et al., 2014; Saito et al., 2018; Skowronski et al., 2016). Therefore, the model may underestimate VE in less vaccinated populations. Underestimation of VE could also occur if unvaccinated people are protected by vaccination in the preceding season. Inference might also be distorted if vaccination has large indirect effects, which our model does not consider.”

6) Please provide a visualisation of the raw data showing how many tests were done in each year, and what was their outcome, stratified by reported vaccination status. How did the control population compare?

We now show the data of all enrolled (i.e., tested) individuals for each season, stratified by age, vaccination status, and test outcome as Figure 1—figure supplement 1.

The proportion of test-positive cases changes between age groups and seasons, consistent with susceptibility of each age group changing over time. The proportion of vaccinated cases among test-positive cases is generally smaller than the proportion of vaccinated cases among test-negative cases, consistent with the protective effect of vaccination.

We also stratified our comorbidity analysis by age, season, vaccination status, and test status (Figure 1—figure supplement 4). Regardless of test outcome, vaccinated people tend to have more high-risk comorbidities than unvaccinated people. We used test-negative cases to adjust for this effect when estimating VE and have clarified this in our overview of the modeling method:

“However, vaccinated individuals may seek medical attention for acute respiratory infection more frequently than non-vaccinees due to correlations between the decision to vaccinate, healthcare-seeking behavior, and underlying medical conditions (Jackson et al., 2005a,b; Belongia et al., 2009). Indeed, we generally observe higher rates of high-risk medical conditions among vaccinated people compared to unvaccinated people (Figure 1—figure supplement 4). We attempted to adjust for this by calculating the fraction of vaccinated people among those who had MAARI and tested negative for influenza (i.e., the test-negative controls, Mathematical expressions for model components: "Vaccination").”

7) Throughout the manuscript the authors dichotomise results into significant/non-significant based on whether p-values are below or above the arbitrary threshold of 0.05 (and in some cases it is only reported whether p-values are above or below, the actual values are not given). While this is common practice, it goes against mainstream statistical thinking which advises against the use of such "bright lines" when reporting or interpreting results (see the recent consensus statement on p-values from the American Statistical Association which "is intended to steer research into a post p<0.05 era." https://www.amstat.org/asa/files/pdfs/P-ValueStatement.pdf). While p-values clearly have a role (though the authors should bear in mind that with enough data, an arbitrarily small difference of no clinical significance can have an arbitrarily small p-value) dichotomising results into significant /non-significant is rarely helpful. We strongly encourage the authors to consider whether this approach is justified in light of the ASA statement and to consider revising the manuscript accordingly.

We agree such dichotomies should be avoided and thank the reviewers for bringing to our attention opportunities for more nuance. For the modelling results, we report 95% confidence intervals for parameter estimates in an effort to focus on effect sizes and windows of uncertainty. Here, we believe that it is reasonable to use terms such as “not significantly different” to denote parameter estimates under different models that cannot be statistically distinguished from each other.

In other places, we have revised the text and figures to report our results more accurately:

Materials and methods text changed from “The G-test of independence was used to determine whether each pair of seasons had significantly different age distributions. We considered differences significant if the Bonferroni-corrected p-value was <0.05” to “The G-test of independence was used to measure differences in seasons' age distributions.”. In Figure 2—figure supplement 1, we now report both p-values and the G-statistic (a measure of effect size) for all pairs of seasons.

We now report the correlation between an age group’s rank in relative risk in a season and its rank in relative size (the fraction of cases) in that season (Figure 2—figure supplement 2). We also report confidence intervals for Pearson’s r instead of p-values.

We removed the dashed line that indicated the value of Spearman’s ρ corresponding to a p-value of 0.05.

We removed p-values and replaced them with 95% confidence intervals of Spearman’s ρ.

8) While not an essential revision, out-of-sample validation of the models (rather than just model comparison) would greatly strengthen the conclusions.

We performed a leave-one-out cross-validation analysis, which yielded the same results. Briefly, we excluded a single season of data, refitted all models to this subset, and evaluated how well each model predicted the birth year distribution of cases for the excluded season (subsection “Evaluation of predictive power”).

Leave-one-out cross-validation gave the same main result as our original findings based on cAIC. The model with the lowest mean-squared prediction error included age-specific medically attended infection risk, age-specific vaccine effectiveness, and HA imprinting by subtype. We have added this to the Results section:

“Our best-fitting model supports subtype-specific imprinting for H1N1 and H3N2 (Figure 5, top row; Appendix 2: Table 1). This model also provides the best predictive power compared to other models in a leave-one-out cross-validation analysis (Figure 5—figure supplement 1; Figure 5—figure supplement 2; Appendix 1: “Evaluation of predictive power”).”